# Biased Temporal Convolution Graph Network for Time Series Forecasting with Missing Values

**Xiaodan Chen**[1]**, Xiucheng Li**[2] (✉)**, Bo Liu**[1]**, Zhijun Li**[1] (✉)
[1] School of Computer Science and Technology, Harbin Institute of Technology
[2] School of Computer Science and Technology, Harbin Institute of Technology (Shenzhen)
`{21B303004@stu., lixiucheng@, 23B936027@stu., lizhijun_os@}hit.edu.cn`

## Abstract

Multivariate time series forecasting plays an important role in various applications ranging from meteorology study, traffic management to economics planning. In the past decades, many efforts have been made toward accurate and reliable forecasting methods development under the assumption of intact input data. However, the time series data from real-world scenarios is often partially observed due to device malfunction or costly data acquisition, which can seriously impede the performance of the existing approaches. A naive employment of imputation methods unavoidably involves error accumulation and leads to suboptimal solutions. Motivated by this, we propose a Biased Temporal Convolution Graph Network that jointly captures the temporal dependencies and spatial structure. In particular, we inject bias into the two carefully developed modules—the Multi-Scale Instance PartialTCN and Biased GCN—to account for missing patterns. The experimental results show that our proposed model is able to achieve up to 9.93% improvements over the existing methods on five real-world benchmark datasets. Our code is available at: https://github.com/chenxiaodanhit/BiTGraph.

## 1 Introduction

Multivariate time series forecasting finds its applications in a wide spectrum of domains such as meteorology, traffic, energy consumption, economics, etc. The real-world demand has spurred the development of various forecasting approaches in the literature. From the generative perspective, the multivariate time series data is produced by a collection of $N$ instances (e.g., sensors) during a period of time. Thus, an accurate characterization of the underlying dynamics requires faithfully modeling both the temporal dependencies (intra-instance correlation) and the spatial structure (inter-instance correlation). The statistical methods—ARIMA (Nelson, 1998), VAR (Zivot & Wang, 2006)—have made early attempts by building autoregressive models to capture the temporal dependencies. However, their linear dependency assumption often leads to poor performance in practice. Inspired by their successes in Natural Language Processing, there has been an increasing trend in designing forecasting models based on RNNs and Transformers to explore their nonlinear modeling and complex pattern extraction capacity (Salinas et al., 2020; Zhou et al., 2021; Liu et al., 2021; Wu et al., 2021; Zhou et al., 2022). Especially, benefiting from the wide-range receptive fields enabled by the attention mechanism, the Transformer-based methods have exhibited excellent prediction performance on long-term forecasting tasks.

Apart from the methods dedicated to temporal dependencies modeling, there is another line of work toward exploiting the spatial correlation of multivariate time series. Many proposals (Salinas et al., 2020; Liu et al., 2021) model the spatial dependencies implicitly and simply rely on a hidden representation to capture the correlation. BRITS (Cao et al., 2018) proposes to use a dense connection layer to learn the correlation between every instance pair, which results in a high model complexity. The advent of graph neural networks (GNNs) (Kipf & Welling, 2016; Defferrard et al., 2016) enables us to effectively explore the non-Euclidean structure data. Indeed, the DCRNN (Li et al., 2018) proposes to build graphs and conduct graph convolution operations to capture the spatial correlation explicitly in traffic flow forecasting, in which the graphs are induced by spatial proximity. To apply

GNNs to the more general forecasting scenarios, in which the graph structures are not available, the proposals (Bai et al., 2020; Wu et al., 2020) propose to learn the graphs adaptively by learning each node an embedding and building the graphs using the node embeddings, which achieves great progress in enhancing the prediction accuracy.

Despite the promising results achieved, the existing methods pay relatively less attention to multivariate time series forecasting with missing values. In practice, the collected time series data is often partially observed, caused by device malfunction, communication failure, or data acquisition difficulty. One commonly adopted solution is to employ the existing time series imputation methods (Cao et al., 2018; Marisca et al., 2022; Cini et al., 2022) and then build the forecasting models on the imputed data. However, this two-step process separates the forecasting from the imputation, and the accumulated errors may impede the model performance and lead to suboptimal solutions. GRU-D (Che et al., 2018) proposes a decayed-GRU mechanism to handle the missing values for time series classification, and a similar idea is also adopted by BRITS (Cao et al., 2018) and GRUI (Luo et al., 2018) for time series imputation. However, these methods are not dedicated to time series forecasting. Neural ODE (Chen et al., 2018) is capable of handling irregularly-sampled time series data and many variants including LatentODE, NeuralCDE, and CRUs (Rubanova et al., 2019; Kidger et al., 2020; Schirmer et al., 2022) have been proposed. However, these methods often entail an ODE-solver computation for each iteration and have to align time steps of different time series, and thus cannot utilize the sparsity of the observations.

Motivated by the above observations, in this paper, we propose a Biased Temporal Convolution Graph Network, dubbed BiTGraph, to jointly capture the temporal dependencies and spatial structure by explicitly exploring the missing values in the model architecture design. We develop two core modules—the Multi-Scale Instance PartialTCN and Biased GCN. The Multi-Scale Instance PartialTCN performs instance-independent partial temporal convolution to capture the intra-instance temporal dependencies contaminated by the missing values. Furthermore, the Biased GCN module explores the spatial structure by constructing a biased graph to account for the missing patterns. Besides, we integrate the two modules with a hierarchical architecture, in which the missing patterns will be updated progressively along the temporal and spatial dimensions to maximize information propagation and minimize the impacts of missing values. To summarize, our contributions are as follows.

- We present BiTGraph to jointly capture the temporal dependencies and spatial structure for the time series forecasting with missing values, the proposed model explicitly considers the missing patterns in its model design.

- We introduce Multi-Scale Instance PartialTCN to effectively model temporal dependencies destroyed by the missing values and present Biased GCN to propagate information among instances by building a biased graph in a missing patterns aware manner.

- BiTGraph achieves up to 9.93% improvements over the existing forecasting methods under various missing values scenarios as verified on five real-world benchmark datasets.

## 2 RELATED WORK

**Time series forecasting with complete data** Due to its practical importance, a lot of efforts have been devoted to developing accurate time series forecasting methods. The classic ARIMA (Nelson, 1998), VAR (Zivot & Wang, 2006) build the autoregressive models based on linear dependency assumption. RNNs-based methods (Salinas et al., 2020) and (Zaremba et al., 2014) exploit the expressive power of recurrent neural networks to relax the linear assumption. Very recently, various Transformer-based methods have been proposed to exploit the wide-range receptive fields of attention mechanism for long-term forecasting. To reduce the quadratic complexity of vanilla attention, Informer (Zhou et al., 2021), Pyraformer (Liu et al., 2021), Autoformer (Wu et al., 2021), and FEDformer (Zhou et al., 2022) have been proposed successively. Non-stationary Transformer (Liu et al., 2022) aims to renovate the attention mechanism to account for the non-stationary property of time series data. PatchTST (Nie et al., 2023) explores the patch and channel-independence design. Apart from enhancing the temporal dynamics modeling capability, many proposals are dedicated to exploring spatial correlation. DCRNN (Li et al., 2018), AGCRN (Bai et al., 2020), MTGNN (Wu et al., 2020), GTS (Shang et al., 2021), and SAGDFN (Jiang et al., 2024) model the spatial structure

with the graph neural networks. In addition, CoST (Woo et al., 2022) and TS2Vec (Yue et al., 2022) approach the time series forecasting from the self-supervised learning perspective.

**Modeling time series with missing values** Caused by device malfunction, communication failure, or costly data acquisition, the real-world collected time series data is often incomplete and partially observed. To fill missing entries, many time series imputation methods—BRITS (Cao et al., 2018), GRIN (Cini et al., 2022), CSDI (Tashiro et al., 2021), SPIN (Marisca et al., 2022), GRIN (Cini et al., 2022), and TIDER (Liu et al., 2023)—have been presented in the machine learning community. To deal with the partially observed time series, one may attempt to build the forecasting models with the imputed results produced by the imputation methods. However, the imputation is disparate from the forecasting in this two-step process, and thus the accumulated errors may seriously degrade the forecasting performance. GRU-D (Che et al., 2018) presents a decayed-GRU to handle the missing values for time series classification without resorting to the imputation. Tang et al. (2020); Zuo et al. (2023) attempt to capture local dependencies based on global statistic characteristics for the missing value forecasting. The neural ODE-based models NeuralCDE, LatentODE, and CRUs (Chen et al., 2018; Rubanova et al., 2019; Schirmer et al., 2022) are capable of handling irregularly-sampled time series data. Nonetheless, they have to align time steps of different time series and cannot utilize the sparsity of the samples.

## 3 PRELIMINARIES

In this paper, we consider the multivariate time series $\mathbf{X} \in \mathbb{R}^{N \times T \times D}$ consisting of $N$ univariate time series $\mathbf{x}^{(1)}$, $\mathbf{x}^{(2)}$, ..., $\mathbf{x}^{(N)}$ collecting over $T$ time steps with $D$-dimension observation. Due to the malfunction of devices, communication failure, or costly data acquisition, there may exist missing values in $\mathbf{X}$, and we use a mask matrix $\mathbf{M} \in \mathbb{R}^{N \times T}$ to represent the missing patterns, which is defined as follows.

$$M_{nt} = \begin{cases} 1, & \text{if } X_{nt} \text{ is observed,} \\ 0, & \text{otherwise,} \end{cases} \tag{1}$$

where $X_{nt}$ denotes the value of $n$-th instance (or channel) at time step $t$, we alternatively use $x_t^{(n)}$ or $X_{nt}$ to represent the same entry. Similarly, we use $\mathbf{m}^{(n)} \in \mathbb{R}^T$ to denote the $n$-th row of the mask matrix $\mathbf{M}$, and both $M_{nt}$ and $m_t^{(n)}$ represent the $n$-row, $t$-column element of $\mathbf{M}$. In addition, the slice notation $\mathbf{x}_{t-H:t} \in \mathbb{R}^{H \times D}$ or $\mathbf{X}_{t-H:t} \in \mathbb{R}^{N \times H \times D}$ denotes the values in a time window of size $H$ from time step $t - H$ to $t - 1$, i.e., the time interval $[t - H, t)$. In the subsequent discussion, we will also refer to the mask as the missing pattern.

**Multivariate time series forecasting with missing values** Given the partial observed multivariate time series $\mathbf{X}$ and the corresponding mask matrix $\mathbf{M}$, the multivariate time series forecasting with missing values problem aims to build a forecasting model $\phi$ to predict the future $F$-step values $\mathbf{Y} = \mathbf{X}_{t:t+F}$ by taking as inputs the historical observation $\mathbf{X}_{t-H:t}$ and its mask $\mathbf{M}_{t-H:t}$, that is, $\hat{\mathbf{Y}} = \phi(\mathbf{X}_{t-H:t}, \mathbf{M}_{t-H:t})$. In the training phase, we only resort to the observed values to provide the learning signals. More formally, the loss function $\mathcal{L}$ of the model can be described as follows.

$$\mathcal{L}(\mathbf{Y}, \hat{\mathbf{Y}}, \mathbf{M}_{t:t+F}) = \frac{\sum_{n=1}^{N} \sum_{\tau=t}^{t+F-1} m_\tau^{(n)} |\hat{y}_\tau^{(n)} - y_\tau^{(n)}|}{\sum_{n=1}^{N} \sum_{\tau=t}^{t+F-1} m_\tau^{(n)}}, \tag{2}$$

which measures the mean absolute error between the predicted values and ground truths.

## 4 METHODOLOGY

The framework of our proposed BiTGraph (Biased Temporal Convolution Graph Network) is shown in Figure 1-(a). It comprises $L$ identical blocks, dubbed Biased TCGBlock (Biased Temporal Convolution Graph Block), which is the basic building block of our proposed method. The Biased TCGBlock consists of two key modules: the Multi-Scale Instance PartialTCN module and the Biased GCN module. The two modules are responsible for fusing the information along the temporal dimension and spatial dimension, respectively. In contrast to the existing time series forecasting methods, we explicitly consider the missing values in the model design and inject bias to account for the different missing patterns, and the model also progressively updates the missing patterns

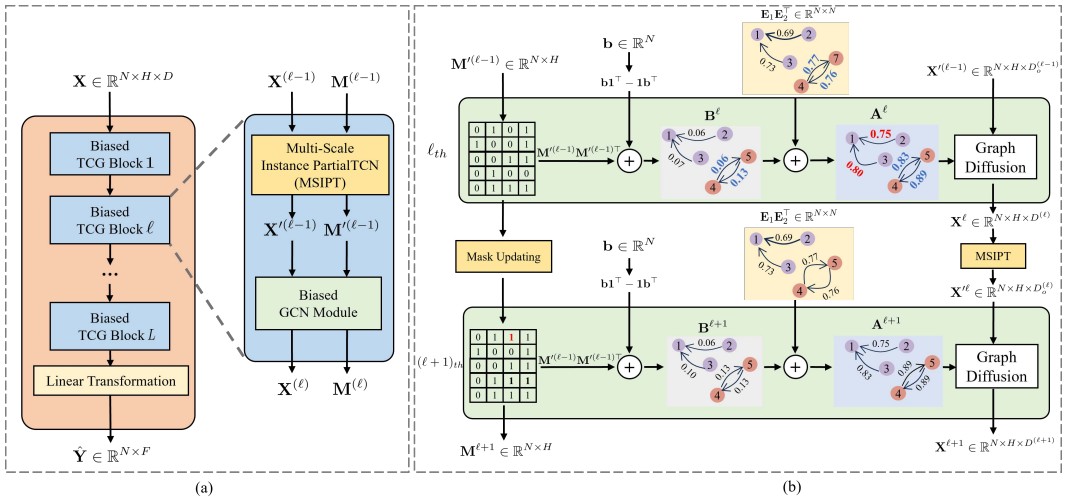

Figure 1: (a) The framework of our proposed BiTGraph, and (b) the illustration of the Biased GCN module.

as the information diffusion proceeds. As the Figure 1-(a) shows, the $\ell$-th block takes as inputs $\mathbf{X}^{(\ell-1)} \in \mathbb{R}^{N \times H \times D^{(\ell-1)}}$ and the missing patterns $\mathbf{M}^{(\ell-1)} \in \mathbb{R}^{N \times H}$, and it produces two transformed tensors $\mathbf{X}^{(\ell)} \in \mathbb{R}^{N \times H \times D^{(\ell)}}$ and $\mathbf{M}^{(\ell)} \in \mathbb{R}^{N \times H}$, where $D^{(\ell)}$ is the feature dimension of the $\ell$-th block.

## 4.1 MULTI-SCALE INSTANCE PARTIALTCN MODULE

In this paper, we opt for Temporal Convolution Network (TCN) as our backbone to capture the temporal dynamics for two main reasons: 1) it has been shown empirically that the TCN exhibits more favorable sequence modeling abilities in comparison to RNNs in a variety of tasks (Bai et al., 2018), 2) the convolution operation permits a simple modification to account for partial observations as evidenced in computer vision (Liu et al., 2018). Different from the vanilla partial convolution (PartialCNN), we propose to apply the partial temporal convolution (PartialTCN) within each time series (instance), i.e., the parameters of PartialTCN are shareable across different instances. The benefits are twofold: 1) we decouple the temporal dependency modeling from spatial correlation modeling, which enables exploring the temporal patterns invariant to instances and enhancing the statistical strengths; 2) the PartialTCN is shareable across instances and this will lead to a more parameter-efficient model, we refer to the resulting approach as Instance ParatialTCN. In addition, we further adopt the multi-scale strategy to develop Multi-Scale Instance ParitalTCN, which can handle missing patterns from different time scales more effectively. Next, we illustrate the module by focusing on a particular instance $\mathbf{x}^{(n)} \in \mathbb{R}^{H \times D}$ and its mask $\mathbf{m}^{(n)} \in \mathbb{R}^{H}$. To keep the notation uncluttered, we drop the upper script temporally.

**Instance PartialTCN** Given the kernel size $K$, the TCN applies the same linear transformation into different time windows under the time translation invariance assumption, i.e.,

$$\mathbf{x}' = \mathbf{x}_{t-K:t} \mathbf{W} + \mathbf{b}, \tag{3}$$

where $\mathbf{x}_{t-K:t} \in \mathbb{R}^{K \times D_i}$ is a time window sequence with $D_i$ input features, $\mathbf{x}' \in \mathbb{R}^{D_o}$ is the output feature map at location $t-1$, and $\mathbf{W} \in \mathbb{R}^{K \times D_i \times D_o}$ and $\mathbf{b} \in \mathbb{R}^{D_o}$ are convolution parameters. Motivated by the success of partial convolutions in vision tasks, we introduce Instance PartialTCN to model the temporal dependencies of partially-observed time series to account for missing values as,

$$\mathbf{x}' = \begin{cases} \frac{K}{\text{sum}(\mathbf{m}_{t-K:t})} \left( \mathbf{x}_{t-K:t} \odot \mathbf{m}_{t-K:t} \right) \mathbf{W} + \mathbf{b}, & \text{if sum}(\mathbf{m}_{t-K:t}) > 0, \\ \mathbf{0}, & \text{otherwise.} \end{cases} \tag{4}$$

where $\odot$ denotes the Hadamard product. The Instance PartialTCN only attends to the time steps with observations to compute the new feature maps and the factor $K / \text{sum}(\mathbf{m}_{t-K:t})$ rescales the

computation result to the same magnitude of convolutions on complete observations. In such a manner, the missing patterns are integrated into the temporal dynamics modeling. As the temporal convolution proceeds, the time steps with missing values will have chances to gather sufficient information from their surrounding neighbors. To account for this, the missing pattern $\mathbf{m}$ is updated as,

$$m_{t-1} = \begin{cases} 1, & \text{if } \text{sum}(\mathbf{m}_{t-K:t}) > 0, \\ 0, & \text{otherwise.} \end{cases} \tag{5}$$

In other words, the time step $t - 1$ is considered filled if we could collect values from the present time window $[t - K, t)$. The missing pattern $\mathbf{m}$ will be progressively filled as the convolution proceeds.

**Multi-Scale Instance PartialTCN** To capture the multi-scale temporal dependencies of the time series, we propose to integrate the multi-scale convolution with different kernel sizes into the Instance PartialTCN, specifically, we adopt $1 \times 3$, $1 \times 5$, and $1 \times 7$ in this paper. Consequently, different kernels will yield multiple different updated missing patterns $\mathbf{m}_i^{(n)} \in \mathbb{R}^H$ for each instance $n$, $1 \leq i \leq N_{\text{ker}}$ and $N_{\text{ker}}$ is the number of kernels. We propose to aggregate these missing patterns generated by different kernels by max pooling as,

$$\mathbf{m}^{(n)} = \max(\mathbf{m}_i^{(n)}), \quad 1 \leq i \leq N_{\text{ker}}. \tag{6}$$

The aggregated $\mathbf{m}^{(n)}$ will then be used in the subsequent graph convolution module to diffuse information along spatial dimensions. By applying the Multi-Scale Instance PartialTCN to each instance $\mathbf{x}^{(n)}$ ($n = 1, 2, \ldots, N$), we transform the input feature map $\mathbf{X}^{(\ell-1)} \in \mathbb{R}^{N \times H \times D_i^{(\ell-1)}}$ and missing pattern $\mathbf{M}^{(\ell-1)} \in \mathbb{R}^{N \times H}$ into $\mathbf{X}'^{(\ell-1)} \in \mathbb{R}^{N \times H \times D_o^{(\ell-1)}}$ and updated missing pattern $\mathbf{M}'^{(\ell-1)} \in \mathbb{R}^{N \times H}$, respectively.

## 4.2 BIASED GCN MODULE

The Multi-Scale Instance PartialTCN focuses on capturing the temporal dynamics hidden in each instance without considering the inter-instance correlation. However, it is equally important to model both the spatial correlation and temporal dependencies for accurate multivariate time series forecasting. In this paper, we propose to use graph convolution networks to explore the spatial structure of the temporally fused feature map $\mathbf{X}'^{(\ell-1)}$ and updated $\mathbf{M}'^{(\ell-1)}$, produced by the Multi-Scale Instance PartialTCN. The graph neural networks have been exploited to model the spatial correlation for time series forecasting in the literature either by using the predefined (Li et al., 2018) or adaptively-learned graph structures (Bai et al., 2020; Wu et al., 2020; Shang et al., 2021), in which each time series is treated as a graph node. In contrast to the existing approaches, we explicitly consider and incorporate a bias term (i.e., prior knowledge) into graph structure learning to account for the missing values, leading to **Biased GCN**. It is therefore able to deliver promising performance in the missing value scenarios. The Biased GCN module is shown in Figure 1-(b).

In this paper, we choose the adaptive graph structure learning approach since it is more flexible and applies to cases where the graph structures are unavailable. In particular, we learn the graph structure or adjacency matrix $\mathbf{A}$ by using two learnable embedding matrices $\mathbf{E}_1, \mathbf{E}_2 \in \mathbb{R}^{N \times D_{\text{node}}}$ as follows.

$$\mathbf{A} = \text{ReLU}(\tanh(\mathbf{E}_1 \mathbf{E}_2^\top)). \tag{7}$$

The $i$-th row of $\mathbf{E}_1$ (resp. $\mathbf{E}_2$), denoted by $\mathbf{e}_i^{(1)}$ (resp. $\mathbf{e}_i^{(2)}$), is the embedding of $i$-th time series and $\mathbf{e}_i^{(1)\top} \mathbf{e}_j^{(2)}$ quantifies the correlation strength from node $i$ to node $j$. The reason we choose two embeddings instead of one $\mathbf{E}$ and computing $\mathbf{A} = \text{ReLU}(\tanh(\mathbf{E}\mathbf{E}^\top))$ is that the spatial correlations are very likely to be asymmetric in practice. This learned adjacency matrix $\mathbf{A}$ will be used by the subsequent graph convolution operation to aggregate information and aid in the eventual forecasting task, and thus the embedding matrices $\mathbf{E}_1$ and $\mathbf{E}_2$ can be learned end-to-end.

However, Eq. 7 fails to account for missing patterns. Intuitively, the information propagation intensity should vary against the missing patterns and we choose the inner product to quantify it as

$$\mathbf{A} = \text{ReLU}(\tanh(\mathbf{E}_1 \mathbf{E}_2^\top)) + \beta \text{softmax}(\mathbf{M}_{t-H:t} \mathbf{M}_{t-H:t}^\top), \tag{8}$$

where the first term denotes global spatial correlation strength indicating the global message passing strengths among nodes, the second term is specific to a particular time window $[t - H, t)$ and can be

considered as a time-window-specific bias that corrects the global message passing strength according to the current missing pattern in graph diffusion process, $\beta$ denotes a learnable global parameter that controls the intensity of the correctness. By intuition, the information propagation should also be directed and more information should flow from nodes with fewer missing values to the ones with more missing values, but the second term is a symmetric matrix and cannot mirror this intuition. To correct this, we assign each node a learnable scalar bias $b_i$ and use $b_i - b_j$ to adjust towards the asymmetries. Let $\mathbf{b} \in \mathbb{R}^N$ be the learnable bias term, we propose to learn the graph structure as,

$$
\begin{aligned}
\mathbf{B} &= \text{softmax}(\mathbf{M}_{t-H:t}\mathbf{M}_{t-H:t}^\top + \mathbf{b}\mathbf{1}^\top - \mathbf{1}\mathbf{b}^\top) \\
\mathbf{A} &= \text{ReLU}(\tanh(\mathbf{E}_1\mathbf{E}_2^\top)) + \beta\mathbf{B},
\end{aligned}
\tag{9}
$$

where $\mathbf{1}$ is a length-$N$ all-one vector. As shown in Figure 1-(b), global message passing strengths between node 4 and node 5 are corrected by the time-window-specific bias.

To ensure the structure sparsity, we clamp the small entries of $\mathbf{A}$ to zeros by only preserving the neighbors of node $i$ with the top-$k$ correlation strengths and use the clamped $\mathbf{A}$ in the graph convolution operation to aggregate information (as will be shown shortly). Being analogous to the Instance PartialTCN, we propose to update the missing patterns of node $i$ after aggregating the information from its spatial neighbors as follows,

$$
\mathbf{m}^{(i)} = \max(\mathbf{m}^{(j)}), \quad j \in \{i\} \cup \mathcal{N}_i,
\tag{10}
$$

where $\mathcal{N}_i$ indicates the neighbors of node $i$ in the graph. The missing pattern updating process is illustrated with the node 1 in Figure 1-(b).

Now considering the $\ell$-th block of the model, it performs the graph convolution to diffuse information as follows.

$$
\mathbf{X}^{(\ell)} = \left(\mathbf{I} + \mathbf{D}_o^{-1}\mathbf{A} + \mathbf{D}_i^{-1}\mathbf{A}^\top\right)\mathbf{X}'^{(\ell-1)}\mathbf{\Theta}^{(\ell)} + \mathbf{b}^{(\ell)},
\tag{11}
$$

where $\mathbf{X}'^{(\ell-1)}$ is the output of the Multi-Scale Instance PartialTCN in the $\ell$-th block, $\mathbf{D}_i$ and $\mathbf{D}_o$ are the in-degree and out-degree matrix of $\mathbf{A}$, respectively, and $\mathbf{\Theta}^{(\ell)}$ and $\mathbf{b}^{(\ell)}$ are the graph convolution parameters of the $\ell$-th block. $\mathbf{X}^{(\ell)}$ and $\mathbf{M}^{(\ell)}$ will then be fed to the next block as the inputs.

## 4.3 HIERARCHICAL ARCHITECTURE

By stacking $L$ layers of Biased TCGBlock, we could enhance both the spatial and temporal receptive fields of the model. We initialize $\mathbf{X}^{(0)}$ and $\mathbf{M}^{(0)}$ with the original partial observation $\mathbf{X} \in \mathbb{R}^{N \times H \times D}$ and its corresponding missing pattern $\mathbf{M} \in \mathbb{R}^{N \times H}$, and the outputs of the $L$-th block are $\mathbf{X}^{(L)} \in \mathbb{R}^{N \times H \times D^{(L)}}$ and $\mathbf{M}^{(L)} \in \mathbb{R}^{N \times H}$. $\mathbf{X}^{(L)}$ fuses both the spatial and temporal features, which will be used to produce the multi-step prediction $\hat{\mathbf{Y}}$ simultaneously by a linear transformation. The mask of $\ell$-th layer $\mathbf{M}^{(\ell)}$ is updated progressively as information flows from bottom to up, and the model parameters are learned by optimizing the prediction loss in Eq. 2.

## 5 EXPERIMENTS

We evaluate BiTGraph against the state-of-the-art forecasting methods under different missing rates on five real-world benchmark datasets. We first assess the forecasting performance of different methods in terms of three commonly used metrics, and then we verify the efficacy of our proposed modules by ablation study.

### 5.1 EXPERIMENT SETTINGS

**Datasets** We select five most commonly used time series forecasting datasets: Metr-LA, Electricity, PEMS, ETTh1, and BeijingAir, whose statistics are summarized in Table 1. The five datasets are collected from different domains and cover diverse magnitude ranges, sampling frequencies, and statistics. We randomly drop the data according to the missing rate $r$ ranging from 0.1 to 0.8, including 0.1, 0.2, 0.4, 0.6, and 0.8.

**Baseline methods** We compare our proposed BiTGraph with the latest state-of-the-art forecasting methods as well as several classic methods. BRITS (Cao et al., 2018), SPIN (Marisca et al., 2022),

Table 1: Dataset description.

|  | **Metr-LA** | **Electricity** | **PEMS** | **ETTh1** | **BeijingAir** |
|---|---|---|---|---|---|
| #Samples ($T$) | 34272 | 26304 | 52116 | 17420 | 8759 |
| #Instances ($N$) | 207 | 321 | 325 | 7 | 36 |
| Frequency | 5 min | 1 h | 5 min | 1 h | 1 h |
| Mean | 53.72 | 2538.79 | 62.62 | 4.58 | 72.01 |
| Variance | 410.53 | $2.26 \times 10^8$ | 92.05 | 42.68 | 79.07 |

GRIN (Cini et al., 2022), GCN-M (Zuo et al., 2023), CRUs (Schirmer et al., 2022) are representative forecasting methods designed specifically for time series with missing values. Meanwhile, we also include three Transformer-based methods, vanilla Transformer (Zerveas et al., 2021), STWA (Cirstea et al., 2022), and FEDformer (Zhou et al., 2022), as well as two Spatial-Temporal GNNs-based methods, AGCRN (Bai et al., 2020) and MTGNN (Wu et al., 2020). Since these five methods require complete input to perform prediction, we study their two variants, namely, filling the missing entries with zeros and the values imputed by TimesNet (Wu et al., 2023), the state-of-the-art time series imputation approach. We denote the corresponding variants as $Model_0$, and $Model_t$, respectively. The missing masks are fed as covariates to guide the forecasting for the latter five baseline methods. The details of baseline methods are presented in Appendix A.

**Implementation details** The number of blocks $L$ of BiTGraph is set to 3, the number of top-$k$ nearest neighbors is set to 10 in all our experiments. The batch size is 32, the learning rate is 0.001. We split the datasets into training, validation, and test datasets with the ratio 0.6/0.2/0.2 chronologically. The future window size $F$ is set to 24 for all methods, and the history window size $H$ for our proposed method is 24. We select the best history window size from the set $\{24, 48, 96\}$ for the baseline methods and report their best results. All methods are trained on Nvidia V100 GPUs. Our method is implemented with PyTorch 2.0 and we use the source codes released by the authors for all baseline methods. We adjust the hyperparameters of baseline methods to obtain the best performance on each dataset, and evaluate the performance of different methods in terms of Mean Absolute Error (MAE), Root Mean Square Error (RMSE), and Mean Absolute Percentage Error (MAPE).

## 5.2 OVERALL PERFORMANCE

Table 2 presents the forecasting performance on the two datasets (Metr-LA and Electricity) of different methods under the missing rates of 0.2, 0.4, 0.6, and 0.8, the results are averaged over five funs[1]. We move the results of the PEMS, ETTh1, and BeijingAir datasets and the results under the missing rate of 0.1 to Appendix B to save space. It can be seen from the table that our proposed BiTGraph is able to achieve the best results in most cases in terms of all three metrics. Its performance gains become more evident when the missing rate grows to 0.8, which benefits from the ability of the Multi-Scale Instance PartialTCN module and the Biased GCN module in handling the missing patterns adaptively. It is worth noting that SPIN and GRIN, both of which are explicitly designed to address missing values, demonstrate a marked superiority. However, their practical applicability is constrained by the necessity of pre-defined graphs. SPIN, $MTGNN_t$, and $STWA_t$ achieves the best results among all baseline methods under different cases. In comparison, our proposed BiTGraph is able to deliver the best results consistently. Notably, it achieves up to 9.93% improvement over the best baseline in terms of RMSE on the Electricity dataset.

## 5.3 ABLATION STUDY

In this section, we conduct ablation studies to evaluate the effectiveness of our proposed modules, Multi-Scale Instance PartialTCN (MSIPT) and Biased GCN (BGCN) modules. The results are shown in Table 3. We divide the MSIPT module or BGCN module into two distinct procedures. The first part (Eq. 6 or Eq. 10) relates to the mask updating process (MUP), whereas the second part (Eq. 4 or Eq. 9) is regarding the information aggregation process (IAP). Firstly, we carry out ablation studies (w/o. MSIPT, w/o. BGCN, and BiTGraph) to assess the joint significance of UID and MUP across temporal

---
[1] The model is trained with five different random seeds.

Table 2: The forecasting performance of different methods.

| Method | Metr-LA | | | Electricity | | |
|---|---|---|---|---|---|---|
| ($r = 0.2$) | MAE | RMSE | MAPE | MAE | RMSE | MAPE |
| BRITS | $8.32 \pm 0.02$ | $13.18 \pm 0.10$ | $18.26 \pm 0.71$ | $1029.30 \pm 1.10$ | $10126.175 \pm 30.57$ | $47.73 \pm 0.35$ |
| SPIN | $6.46 \pm 0.07$ | $11.21 \pm 0.05$ | $12.98 \pm 0.03$ | – | – | – |
| GRIN | $6.80 \pm 0.02$ | $12.24 \pm 0.12$ | $16.18 \pm 0.24$ | – | – | – |
| GCN-M | $6.78 \pm 0.03$ | $11.12 \pm 0.04$ | $13.50 \pm 0.02$ | – | – | – |
| CRUs | $10.80 \pm 0.02$ | $12.49 \pm 0.15$ | $19.66 \pm 0.54$ | $464.66 \pm 4.14$ | $5276.49 \pm 53.36$ | $\mathbf{25.64 \pm 0.53}$ |
| $AGCRN_0$ | $14.88 \pm 0.05$ | $14.21 \pm 0.04$ | $28.94 \pm 0.07$ | $1307.62 \pm 4.53$ | $13217.78 \pm 26.81$ | $62.65 \pm 0.22$ |
| $Transformer_0$ | $7.14 \pm 0.06$ | $13.08 \pm 0.08$ | $17.07 \pm 0.09$ | $296.03 \pm 5.77$ | $2432.09 \pm 22.15$ | $29.14 \pm 0.27$ |
| $FEDformer_0$ | $7.09 \pm 0.03$ | $12.75 \pm 0.14$ | $16.73 \pm 0.19$ | $368.29 \pm 3.71$ | $2574.37 \pm 25.89$ | $31.29 \pm 0.31$ |
| $STWA_0$ | $6.24 \pm 0.07$ | $10.99 \pm 0.11$ | $12.89 \pm 0.13$ | $272.60 \pm 7.35$ | $2263.55 \pm 24.10$ | $28.52 \pm 0.26$ |
| $MTGNN_0$ | $6.34 \pm 0.07$ | $10.96 \pm 0.10$ | $12.51 \pm 0.19$ | $274.68 \pm 5.56$ | $\underline{2016.44 \pm 13.77}$ | $28.54 \pm 0.19$ |
| $AGCRN_t$ | $13.72 \pm 0.06$ | $13.11 \pm 0.23$ | $27.06 \pm 0.18$ | $1049.23 \pm 12.06$ | $11751.49 \pm 20.67$ | $57.76 \pm 0.18$ |
| $Transformer_t$ | $6.90 \pm 0.08$ | $12.98 \pm 0.13$ | $16.49 \pm 0.21$ | $280.12 \pm 6.78$ | $2274.28 \pm 25.18$ | $28.74 \pm 0.35$ |
| $FEDformer_t$ | $6.89 \pm 0.06$ | $11.75 \pm 0.17$ | $16.01 \pm 0.09$ | $313.59 \pm 4.96$ | $2666.93 \pm 26.31$ | $32.83 \pm 0.23$ |
| $STWA_t$ | $6.20 \pm 0.02$ | $\underline{10.71 \pm 0.11}$ | $12.26 \pm 0.18$ | $\underline{261.92 \pm 4.65}$ | $2089.65 \pm 19.35$ | $\underline{27.37 \pm 0.26}$ |
| $MTGNN_t$ | $\underline{6.13 \pm 0.02}$ | $10.76 \pm 0.07$ | $\underline{12.11 \pm 0.19}$ | $269.25 \pm 5.27$ | $2175.24 \pm 12.49$ | $27.71 \pm 0.78$ |
| BiTGraph | $\mathbf{6.04 \pm 0.02}$ | $\mathbf{10.69 \pm 0.02}$ | $\mathbf{11.69 \pm 0.11}$ | $\mathbf{243.23 \pm 2.12}$ | $\mathbf{1834.18 \pm 15.36}$ | $27.38 \pm 0.46$ |

| Method | Metr-LA | | | Electricity | | |
|---|---|---|---|---|---|---|
| ($r = 0.4$) | MAE | RMSE | MAPE | MAE | RMSE | MAPE |
| BRITS | $8.38 \pm 0.08$ | $12.97 \pm 0.11$ | $18.39 \pm 0.28$ | $1029.73 \pm 1.48$ | $10136.39 \pm 63.63$ | $47.96 \pm 0.56$ |
| SPIN | $6.52 \pm 0.07$ | $11.94 \pm 0.41$ | $13.22 \pm 1.00$ | – | – | – |
| GRIN | $6.91 \pm 0.09$ | $12.60 \pm 0.21$ | $16.59 \pm 0.18$ | – | – | – |
| GCN-M | $7.09 \pm 0.01$ | $12.42 \pm 0.03$ | $17.06 \pm 0.04$ | – | – | – |
| CRUs | $10.94 \pm 0.08$ | $13.18 \pm 0.44$ | $20.13 \pm 0.23$ | $496.95 \pm 6.03$ | $5397.31 \pm 52.52$ | $27.94 \pm 0.33$ |
| $AGCRN_0$ | $14.87 \pm 0.04$ | $14.30 \pm 0.09$ | $29.92 \pm 0.06$ | $1526.90 \pm 13.77$ | $14823.39 \pm 21.68$ | $68.73 \pm 0.41$ |
| $Transformer_0$ | $7.25 \pm 0.04$ | $12.97 \pm 0.06$ | $17.72 \pm 0.08$ | $310.88 \pm 4.67$ | $2586.69 \pm 22.73$ | $31.79 \pm 0.15$ |
| $FEDformer_0$ | $7.15 \pm 0.02$ | $12.89 \pm 0.07$ | $16.91 \pm 0.12$ | $406.17 \pm 8.91$ | $3606.49 \pm 27.73$ | $33.14 \pm 0.33$ |
| $STWA_0$ | $6.37 \pm 0.05$ | $11.19 \pm 0.06$ | $13.13 \pm 0.16$ | $292.47 \pm 4.64$ | $2764.34 \pm 20.06$ | $29.07 \pm 0.16$ |
| $MTGNN_0$ | $6.34 \pm 0.05$ | $11.10 \pm 0.03$ | $12.79 \pm 0.08$ | $305.46 \pm 6.77$ | $2576.44 \pm 25.51$ | $\underline{23.15 \pm 0.37}$ |
| $AGCRN_t$ | $12.73 \pm 0.02$ | $12.49 \pm 0.14$ | $24.13 \pm 0.16$ | $1283.27 \pm 8.49$ | $13743.42 \pm 49.38$ | $58.62 \pm 0.36$ |
| $Transformer_t$ | $6.99 \pm 0.06$ | $12.49 \pm 0.13$ | $16.45 \pm 0.08$ | $300.43 \pm 10.17$ | $2529.26 \pm 19.14$ | $28.86 \pm 0.20$ |
| $FEDformer_t$ | $7.10 \pm 0.05$ | $12.63 \pm 0.13$ | $16.62 \pm 0.06$ | $330.90 \pm 7.76$ | $2711.30 \pm 22.31$ | $29.24 \pm 0.18$ |
| $STWA_t$ | $6.28 \pm 0.03$ | $10.93 \pm 0.14$ | $12.68 \pm 0.07$ | $289.59 \pm 6.13$ | $2355.34 \pm 17.67$ | $28.29 \pm 0.31$ |
| $MTGNN_t$ | $\underline{6.26 \pm 0.05}$ | $\underline{10.90 \pm 0.10}$ | $\underline{12.49 \pm 0.04}$ | $\underline{281.32 \pm 6.82}$ | $\underline{2236.74 \pm 16.81}$ | $28.46 \pm 0.19$ |
| BiTGraph | $\mathbf{6.13 \pm 0.01}$ | $\mathbf{10.76 \pm 0.02}$ | $\mathbf{12.41 \pm 0.12}$ | $270.14 \pm 3.77$ | $2091.88 \pm 30.49$ | $\mathbf{22.04 \pm 0.36}$ |

| Method | Metr-LA | | | Electricity | | |
|---|---|---|---|---|---|---|
| ($r = 0.6$) | MAE | RMSE | MAPE | MAE | RMSE | MAPE |
| BRITS | $8.48 \pm 0.02$ | $12.94 \pm 0.08$ | $18.66 \pm 0.22$ | $1029.38 \pm 1.84$ | $10118.18 \pm 33.04$ | $48.25 \pm 0.29$ |
| SPIN | $6.61 \pm 0.02$ | $11.35 \pm 0.17$ | $13.31 \pm 0.12$ | – | – | – |
| GRIN | $7.04 \pm 0.04$ | $12.71 \pm 0.14$ | $17.04 \pm 0.03$ | – | – | – |
| GCN-M | $7.27 \pm 0.02$ | $11.55 \pm 0.02$ | $16.42 \pm 0.03$ | – | – | – |
| CRUs | $11.02 \pm 0.02$ | $13.38 \pm 0.24$ | $20.40 \pm 0.04$ | $664.07 \pm 9.88$ | $8126.82 \pm 59.42$ | $31.44 \pm 0.45$ |
| $AGCRN_0$ | $14.87 \pm 0.04$ | $14.30 \pm 0.09$ | $29.92 \pm 0.06$ | $1945.61 \pm 6.38$ | $13891.03 \pm 17.38$ | $75.20 \pm 0.29$ |
| $Transformer_0$ | $7.46 \pm 0.01$ | $12.03 \pm 0.05$ | $17.09 \pm 0.07$ | $346.43 \pm 5.59$ | $2952.28 \pm 25.54$ | $28.96 \pm 0.37$ |
| $FEDformer_0$ | $7.50 \pm 0.04$ | $12.32 \pm 0.03$ | $17.31 \pm 0.07$ | $535.72 \pm 7.67$ | $5329.18 \pm 26.71$ | $42.09 \pm 0.46$ |
| $STWA_0$ | $6.82 \pm 0.02$ | $11.72 \pm 0.10$ | $13.66 \pm 0.04$ | $325.47 \pm 6.62$ | $2479.75 \pm 21.17$ | $30.06 \pm 0.22$ |
| $MTGNN_0$ | $6.95 \pm 0.03$ | $12.09 \pm 0.02$ | $13.87 \pm 0.09$ | $329.18 \pm 4.61$ | $2490.45 \pm 23.38$ | $28.20 \pm 0.27$ |
| $AGCRN_t$ | $12.73 \pm 0.02$ | $12.49 \pm 0.14$ | $24.13 \pm 0.16$ | $1374.64 \pm 7.11$ | $12069.56 \pm 19.73$ | $61.92 \pm 0.28$ |
| $Transformer_t$ | $7.22 \pm 0.08$ | $13.61 \pm 0.17$ | $16.75 \pm 0.06$ | $327.17 \pm 8.68$ | $2506.82 \pm 23.17$ | $29.27 \pm 0.40$ |
| $FEDformer_t$ | $7.26 \pm 0.04$ | $13.08 \pm 0.07$ | $17.16 \pm 0.03$ | $341.66 \pm 5.25$ | $2682.73 \pm 24.97$ | $29.87 \pm 0.09$ |
| $STWA_t$ | $\underline{6.55 \pm 0.02}$ | $11.28 \pm 0.07$ | $13.57 \pm 0.03$ | $312.25 \pm 5.36$ | $2407.39 \pm 23.05$ | $29.05 \pm 0.11$ |
| $MTGNN_t$ | $6.63 \pm 0.02$ | $\underline{11.10 \pm 0.04}$ | $13.48 \pm 0.05$ | $\underline{309.59 \pm 4.73}$ | $\underline{2399.51 \pm 20.09}$ | $\mathbf{25.37 \pm 0.13}$ |
| BiTGraph | $\mathbf{6.32 \pm 0.01}$ | $\mathbf{10.93 \pm 0.03}$ | $\mathbf{12.67 \pm 0.11}$ | $295.23 \pm 2.75$ | $2239.06 \pm 26.39$ | $\underline{27.38 \pm 0.49}$ |

| Method | Metr-LA | | | Electricity | | |
|---|---|---|---|---|---|---|
| ($r = 0.8$) | MAE | RMSE | MAPE | MAE | RMSE | MAPE |
| BRITS | $8.56 \pm 0.09$ | $13.03 \pm 0.18$ | $18.92 \pm 0.09$ | $1027.28 \pm 0.50$ | $10150.54 \pm 31.05$ | $48.04 \pm 0.02$ |
| SPIN | $6.68 \pm 0.31$ | $11.42 \pm 0.35$ | $14.41 \pm 1.20$ | – | – | – |
| GRIN | $8.00 \pm 0.02$ | $12.68 \pm 0.09$ | $18.35 \pm 0.05$ | – | – | – |
| GCN-M | $7.75 \pm 0.03$ | $11.65 \pm 0.04$ | $17.94 \pm 0.02$ | – | – | – |
| CRUs | $11.35 \pm 0.12$ | $14.06 \pm 0.70$ | $22.08 \pm 0.22$ | $623.63 \pm 13.07$ | $7033.29 \pm 17.85$ | $33.29 \pm 0.74$ |
| $AGCRN_0$ | $14.86 \pm 0.01$ | $14.27 \pm 0.02$ | $29.92 \pm 0.08$ | $2351.41 \pm 26.79$ | $16824.28 \pm 29.33$ | $207.77 \pm 0.56$ |
| $Transformer_0$ | $8.06 \pm 0.02$ | $12.82 \pm 0.05$ | $18.37 \pm 0.11$ | $398.99 \pm 6.62$ | $3612.37 \pm 24.19$ | $30.07 \pm 0.18$ |
| $FEDformer_0$ | $7.83 \pm 0.05$ | $12.97 \pm 0.14$ | $17.93 \pm 0.06$ | $676.93 \pm 5.62$ | $7859.76 \pm 31.13$ | $64.79 \pm 0.35$ |
| $STWA_0$ | $7.57 \pm 0.06$ | $12.15 \pm 0.07$ | $17.31 \pm 0.12$ | $376.26 \pm 5.36$ | $3512.37 \pm 22.09$ | $31.15 \pm 0.08$ |
| $MTGNN_0$ | $7.45 \pm 0.03$ | $12.21 \pm 0.08$ | $17.22 \pm 0.09$ | $383.89 \pm 6.72$ | $3539.74 \pm 15.22$ | $30.29 \pm 0.11$ |
| $AGCRN_t$ | $14.88 \pm 0.01$ | $14.20 \pm 0.05$ | $29.92 \pm 0.10$ | $1841.76 \pm 6.87$ | $17376.51 \pm 44.79$ | $70.38 \pm 0.56$ |
| $Transformer_t$ | $7.32 \pm 0.04$ | $12.96 \pm 0.08$ | $16.87 \pm 0.05$ | $391.83 \pm 4.17$ | $3451.33 \pm 5.62$ | $32.26 \pm 0.17$ |
| $FEDformer_t$ | $7.33 \pm 0.08$ | $13.17 \pm 0.06$ | $16.71 \pm 0.04$ | $380.06 \pm 3.39$ | $3335.18 \pm 20.10$ | $31.56 \pm 0.13$ |
| $STWA_t$ | $6.90 \pm 0.03$ | $11.30 \pm 0.05$ | $13.69 \pm 0.07$ | $362.25 \pm 3.21$ | $3156.68 \pm 24.41$ | $29.22 \pm 0.15$ |
| $MTGNN_t$ | $\underline{6.79 \pm 0.04}$ | $\mathbf{11.05 \pm 0.07}$ | $\underline{13.54 \pm 0.10}$ | $\underline{355.68 \pm 5.11}$ | $\underline{3023.30 \pm 11.46}$ | $\underline{28.78 \pm 0.31}$ |
| BiTGraph | $\mathbf{6.63 \pm 0.01}$ | $\underline{11.20 \pm 0.00}$ | $\mathbf{13.44 \pm 0.02}$ | $\mathbf{347.35 \pm 1.76}$ | $\mathbf{2839.79 \pm 25.49}$ | $\mathbf{27.97 \pm 0.27}$ |

Table 3: The results of ablation studies on Metr, Electricity, and PEMS datasets under the missing rates of 0.2, 0.4, and 0.6.

| Missing Rate | Model | Metr | | Electricity | | PEMS | |
|---|---|---|---|---|---|---|---|
| | | MAE | RMSE | MAE | RMSE | MAE | RMSE |
| 0.2 | TCGNet | $6.25 \pm 0.01$ | $11.01 \pm 0.02$ | $255.46 \pm 2.86$ | $2026.43 \pm 35.79$ | $1.94 \pm 0.00$ | $4.35 \pm 0.02$ |
| | w/o. MSIPT | $6.34 \pm 0.04$ | $11.25 \pm 0.05$ | $279.73 \pm 4.91$ | $2159.16 \pm 28.74$ | $1.97 \pm 0.02$ | $4.40 \pm 0.04$ |
| | w/o. BGCN | $6.26 \pm 0.03$ | $11.33 \pm 0.03$ | $263.69 \pm 3.32$ | $2029.16 \pm 25.56$ | $1.94 \pm 0.00$ | $4.37 \pm 0.03$ |
| | w/o. Eq. 9 | $6.14 \pm 0.02$ | $\underline{10.72 \pm 0.02}$ | $250.75 \pm 5.01$ | $2020.13 \pm 36.75$ | $1.94 \pm 0.02$ | $4.33 \pm 0.03$ |
| | w/o. Eq. 4 | $\underline{6.12 \pm 0.01}$ | $10.91 \pm 0.03$ | $\underline{246.18 \pm 2.89}$ | $\underline{2003.22 \pm 35.09}$ | $\underline{1.93 \pm 0.01}$ | $\underline{4.30 \pm 0.02}$ |
| | BiTGraph | $\mathbf{6.04 \pm 0.02}$ | $\mathbf{10.69 \pm 0.02}$ | $\mathbf{243.23 \pm 2.12}$ | $\mathbf{1834.18 \pm 15.36}$ | $\mathbf{1.90 \pm 0.01}$ | $\mathbf{4.28 \pm 0.01}$ |
| 0.4 | TCGNet | $6.41 \pm 0.03$ | $11.14 \pm 0.04$ | $284.39 \pm 5.26$ | $2323.03 \pm 40.15$ | $1.99 \pm 0.01$ | $4.47 \pm 0.03$ |
| | w/o. MSIPT | $6.48 \pm 0.02$ | $11.20 \pm 0.03$ | $299.34 \pm 4.17$ | $2361.79 \pm 37.82$ | $2.02 \pm 0.02$ | $4.50 \pm 0.03$ |
| | w/o. BGCN | $6.40 \pm 0.02$ | $11.20 \pm 0.02$ | $291.81 \pm 3.87$ | $2337.69 \pm 31.98$ | $1.98 \pm 0.01$ | $4.45 \pm 0.02$ |
| | w/o. Eq. 9 | $\underline{6.18 \pm 0.01}$ | $\underline{10.81 \pm 0.04}$ | $282.12 \pm 2.88$ | $2236.82 \pm 30.26$ | $1.98 \pm 0.01$ | $\mathbf{4.32 \pm 0.02}$ |
| | w/o. Eq. 4 | $6.25 \pm 0.00$ | $10.87 \pm 0.02$ | $280.30 \pm 2.73$ | $2277.50 \pm 28.49$ | $1.97 \pm 0.01$ | $\underline{4.34 \pm 0.01}$ |
| | BiTGraph | $\mathbf{6.13 \pm 0.01}$ | $\mathbf{10.76 \pm 0.02}$ | $\mathbf{270.14 \pm 3.77}$ | $\mathbf{2091.88 \pm 30.49}$ | $\mathbf{1.96 \pm 0.00}$ | $\underline{4.34 \pm 0.02}$ |
| 0.6 | TCGNet | $6.48 \pm 0.02$ | $11.10 \pm 0.05$ | $313.60 \pm 3.29$ | $2372.36 \pm 36.19$ | $2.04 \pm 0.02$ | $4.55 \pm 0.02$ |
| | w/o. MSIPT | $6.65 \pm 0.03$ | $11.50 \pm 0.04$ | $332.39 \pm 3.82$ | $2469.15 \pm 33.63$ | $2.09 \pm 0.02$ | $4.62 \pm 0.04$ |
| | w/o. BGCN | $6.65 \pm 0.02$ | $11.94 \pm 0.01$ | $322.68 \pm 2.74$ | $2487.22 \pm 25.39$ | $\underline{2.03 \pm 0.00}$ | $\underline{4.49 \pm 0.01}$ |
| | w/o. Eq. 9 | $\underline{6.35 \pm 0.03}$ | $11.06 \pm 0.02$ | $308.59 \pm 3.97$ | $2366.39 \pm 32.16$ | $\underline{2.03 \pm 0.01}$ | $4.52 \pm 0.02$ |
| | w/o. Eq. 4 | $6.38 \pm 0.02$ | $\mathbf{10.84 \pm 0.02}$ | $301.25 \pm 2.05$ | $2312.39 \pm 22.46$ | $\underline{2.03 \pm 0.02}$ | $4.54 \pm 0.00$ |
| | BiTGraph | $\mathbf{6.32 \pm 0.01}$ | $\underline{10.93 \pm 0.03}$ | $\mathbf{295.23 \pm 2.75}$ | $\mathbf{2239.06 \pm 26.39}$ | $\mathbf{1.99 \pm 0.01}$ | $\mathbf{4.47 \pm 0.01}$ |

and spatial dimensions. The results reveal that when we adopt MSIPT or BGCN the performance drops significantly, which can be explained by that the MUP builds a complete information-passing path between the spatial and temporal dimensions and the absence of any module will cut off the information flow between the two dimensions.

To further validate the effectiveness of the IAP and MUP, we conduct ablation studies by modifying the IAP. First, we replace the temporal convolution in the MSIPT module with the standard convolution operation (e.g., Eq. 3). Next, we alter the generation of adjacent matrix $A$ by using Eq. 7. The results are shown in the fourth and fifth rows of the table. As we can see, the replacement of either spatial or temporal operations within IAP leads to a notable performance drop, which further verifies the effectiveness of our proposed modules in handling missing values. We also conduct the corresponding ablation study under the block missing scenarios and the results are shown in Appendix C. The results of parameter sensitivity including window size $H$, the number of blocks $L$, and the number of nearest neighbors $k$ are shown in Appendix D. We analyze the role of $\beta$ in Appendix E. We visualize the prediction curves in Appendix F. The model complexity analysis is given in Appendix G.

## 6 CONCLUSIONS

In this paper, we present BiTGraph for the time series forecasting with missing values. BiTGraph jointly captures the temporal dynamics and spatial structure by explicitly taking the missing values into consideration. We inject bias into the two carefully designed modules, the Multi-Scale Instance PartialTCN and Biased GCN, to account for the missing patterns. The experimental results on five real-world benchmark datasets verify its superiority under various missing value scenarios. The ablation studies also show that its excellent performance stems from the two carefully designed Multi-Scale Instance PartialTCN and Biased GCN components. In the future, we would like to explore the Transformer architecture as the backbone of our temporal module to further enhance its long-term forecasting performance for partially observed time series data.

ACKNOWLEDGEMENTS

This work is supported by the National Natural Science Foundation of China under Grant No. 62206074 and Grant No. 62072137, Shenzhen College Stability Support Plan under Grant No. GXWD20220811173233001, and the National Key R&D Program of China under Grant No. 2023YFB4503100.

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

# A EXPERIMENTAL DETAILS

## A.1 DETAILS OF BASELINE MODELS

The details of the baseline models are briefly summarized as follows. For BRITS, SPIN, GRIN, CRUs, AGCRN, MTGNN, FEDformer, and STWA, we use the source codes released by their authors. While for Transformer, we use the version implemented in FEDformer.

- BRITS[2]: It is a time series imputation model that combines a Bidirectional Recurrent Neural Network with the time decay mechanism to establish the relationship between missing values and observed data.

- SPIN[3]: It handles missing values by constructing a sparsely connected graph in both spatial and temporal dimensions.

- GRIN[4]: It incorporates recurrent neural network (RNN) and graph neural network (GNN) to capture inter- and intra-series dependencies to build the relationship between missing values and observed ones.

- GCN-M[5]: It considers local spatiotemporal features and global historical patterns in an attention-based memory network.

- CRUs[6]: It combines the Kalman filter and encoder-decoder frameworks to update the continuous hidden states.

- AGCRN[7]: The adaptive graph and node-specific patterns are learned by node embeddings and matrix factorization, respectively.

- MTGNN[8]: The approach constructs a skew-symmetrical spatial correlation matrix and employs the Temporal Convolutional Network (TCN) and GCN to capture the intra- and inter-series dependencies.

- Transformer: The classic sequential model that uses a stack of self-attention blocks to capture the temporal dependencies in time series.

- FEDformer[9]: The method decomposes the time series into seasonal and trend components and employs the self-attention mechanism in the frequency domain.

- STWA[10]: It constructs spatial-temporal aware embeddings within the self-attention mechanism and introduces window attention to reduce complexity.

## A.2 DETAILS OF METRICS

The metrics of mean absolute error (MAE), root mean square error (RMSE), and mean absolute percentage error (MAPE) adopted in the paper are defined as follows.

$$\text{MAE} = \frac{\sum_{ij \in \Omega} |y_{ij} - \hat{y}_{ij}|}{|\Omega|}, \text{RMSE} = \sqrt{\frac{\sum_{ij \in \Omega} (y_{ij} - \hat{y}_{ij})^2}{|\Omega|}}, \text{MAPE} = \sum_{ij \in \Omega} \frac{|y_{ij} - \hat{y}_{ij}|}{|\Omega| \cdot |y_{ij}|} \quad (12)$$

where $\Omega$ represents the index set of observed values.

---

[2] https://github.com/caow13/BRITS
[3] https://github.com/Graph-Machine-Learning-Group/spin
[4] https://github.com/Graph-Machine-Learning-Group/grin
[5] https://github.com/JingweiZuo/GCN-M
[6] https://github.com/boschresearch/Continuous-Recurrent-Units
[7] https://github.com/LeiBAI/AGCRN
[8] https://github.com/nnzhan/MTGNN
[9] https://github.com/DAMO-DI-ML/ICML2022-FEDformer
[10] https://github.com/razvanc92/ST-WA

Table 4: The results of forecasting error on Metr-LA and Electricity datasets with the missing rate $r = 0.1$.

| Method | Metr-LA | | | Electricity | | |
|---|---|---|---|---|---|---|
| ($r = 0.1$) | MAE | RMSE | MAPE | MAE | RMSE | MAPE |
| BRITS | $8.21 \pm 0.01$ | $12.63 \pm 0.10$ | $18.04 \pm 0.20$ | $1027.47 \pm 3.92$ | $10157.39 \pm 64.012$ | $47.54 \pm 0.01$ |
| SPIN | $6.36 \pm 0.00$ | $11.07 \pm 0.01$ | $12.00 \pm 0.00$ | – | – | – |
| GRIN | $6.69 \pm 0.02$ | $12.27 \pm 0.00$ | $15.88 \pm 0.01$ | – | – | – |
| GCN-M | $6.72 \pm 0.01$ | $12.33 \pm 0.01$ | $13.06 \pm 0.02$ | – | – | – |
| CRU | $10.51 \pm 0.03$ | $13.00 \pm 0.63$ | $19.71 \pm 0.31$ | $334.40 \pm 31.80$ | $2923.44 \pm 39.73$ | $24.99 \pm 0.72$ |
| $\text{AGCRN}_0$ | $14.66 \pm 0.01$ | $14.00 \pm 0.02$ | $29.30 \pm 0.13$ | $1361.11 \pm 8.39$ | $12569.27 \pm 30.09$ | $62.54 \pm 0.31$ |
| $\text{Transformer}_0$ | $7.07 \pm 0.05$ | $12.97 \pm 0.08$ | $16.78 \pm 0.13$ | $289.65 \pm 3.39$ | $2296.17 \pm 27.77$ | $25.08 \pm 0.32$ |
| $\text{FEDformer}_0$ | $6.96 \pm 0.03$ | $12.37 \pm 0.08$ | $16.22 \pm 0.14$ | $337.16 \pm 5.25$ | $2713.72 \pm 33.37$ | $30.24 \pm 0.28$ |
| $\text{STWA}_0$ | $6.22 \pm 0.04$ | $14.64 \pm 0.06$ | $12.71 \pm 0.05$ | $269.81 \pm 5.95$ | $2039.64 \pm 24.06$ | $22.39 \pm 0.33$ |
| $\text{MTGNN}_0$ | $6.25 \pm 0.06$ | $10.68 \pm 0.07$ | $12.18 \pm 0.05$ | $256.98 \pm 5.12$ | $1974.50 \pm 14.69$ | $20.95 \pm 0.26$ |
| $\text{AGCRN}_t$ | $13.72 \pm 0.06$ | $13.11 \pm 0.23$ | $27.06 \pm 0.18$ | $1109.57 \pm 3.95$ | $10794.08 \pm 34.26$ | $57.95 \pm 0.18$ |
| $\text{Transformer}_t$ | $6.90 \pm 0.07$ | $12.81 \pm 0.04$ | $16.49 \pm 0.06$ | $265.76 \pm 6.03$ | $2064.82 \pm 23.51$ | $20.06 \pm 0.27$ |
| $\text{FEDformer}_t$ | $6.61 \pm 0.03$ | $11.09 \pm 0.10$ | $13.23 \pm 0.06$ | $283.63 \pm 2.15$ | $2269.11 \pm 19.86$ | $23.23 \pm 0.18$ |
| $\text{STWA}_t$ | $6.17 \pm 0.08$ | $10.82 \pm 0.09$ | $12.14 \pm 0.04$ | $248.87 \pm 3.79$ | $1945.16 \pm 23.20$ | $18.20 \pm 0.16$ |
| $\text{MTGNN}_t$ | $\underline{6.10 \pm 0.02}$ | $\mathbf{10.69 \pm 0.03}$ | $\underline{12.02 \pm 0.08}$ | $254.67 \pm 3.22$ | $1994.07 \pm 25.99$ | $22.41 \pm 0.26$ |
| BiTGraph | $\mathbf{5.96 \pm 0.01}$ | $\underline{10.71 \pm 0.00}$ | $\mathbf{11.13 \pm 0.02}$ | $\mathbf{231.70 \pm 1.76}$ | $\mathbf{1823.18 \pm 25.49}$ | $\mathbf{17.93 \pm 0.27}$ |

Table 5: The results of forecasting error on PEMS, ETTh1, and BeijingAir datasets with the missing rate $r = 0.1$ and 0.8.

| Method | PEMS | | ETTh1 | | BeijingAir | |
|---|---|---|---|---|---|---|
| (0.1) | MAE | RMSE | MAE | RMSE | MAE | RMSE |
| BRITS | $3.06 \pm 0.01$ | $6.39 \pm 0.02$ | $1.76 \pm 0.03$ | $3.17 \pm 0.03$ | $45.79 \pm 0.24$ | $67.83 \pm 0.21$ |
| SPIN | $2.03 \pm 0.00$ | $4.62 \pm 0.00$ | – | – | $44.93 \pm 0.03$ | $68.13 \pm 0.75$ |
| GRIN | $2.63 \pm 0.01$ | $6.03 \pm 0.06$ | – | – | $45.96 \pm 0.25$ | $67.06 \pm 1.30$ |
| GCNM | $2.13 \pm 0.02$ | $5.29 \pm 0.07$ | – | – | $47.68 \pm 0.11$ | $68.29 \pm 0.09$ |
| CRUs | $3.21 \pm 0.02$ | $6.03 \pm 0.01$ | $2.80 \pm 0.03$ | $4.82 \pm 0.05$ | $56.92 \pm 0.72$ | $76.13 \pm 0.87$ |
| $\text{AGCRN}_0$ | $5.10 \pm 0.07$ | $10.07 \pm 0.06$ | $2.39 \pm 0.04$ | $4.76 \pm 0.08$ | $55.50 \pm 0.12$ | $81.31 \pm 0.19$ |
| $\text{Transformer}_0$ | $2.75 \pm 0.07$ | $6.15 \pm 0.02$ | $1.88 \pm 0.05$ | $3.25 \pm 0.07$ | $48.58 \pm 0.07$ | $69.50 \pm 0.21$ |
| $\text{FEDformer}_0$ | $2.61 \pm 0.05$ | $5.76 \pm 0.10$ | $1.69 \pm 0.02$ | $3.22 \pm 0.04$ | $49.65 \pm 0.09$ | $72.77 \pm 0.13$ |
| $\text{STWA}_0$ | $2.01 \pm 0.04$ | $4.57 \pm 0.03$ | $1.75 \pm 0.00$ | $3.15 \pm 0.00$ | $46.71 \pm 0.14$ | $70.10 \pm 0.07$ |
| $\text{MTGNN}_0$ | $2.02 \pm 0.01$ | $4.52 \pm 0.04$ | $1.58 \pm 0.01$ | $2.99 \pm 0.02$ | $44.37 \pm 0.05$ | $65.92 \pm 0.06$ |
| $\text{AGCRN}_t$ | $5.08 \pm 0.02$ | $10.05 \pm 0.00$ | $2.16 \pm 0.03$ | $4.29 \pm 0.05$ | $47.08 \pm 0.26$ | $69.62 \pm 0.37$ |
| $\text{Transformer}_t$ | $2.54 \pm 0.02$ | $6.05 \pm 0.03$ | $1.72 \pm 0.01$ | $3.25 \pm 0.13$ | $47.43 \pm 0.16$ | $69.69 \pm 0.08$ |
| $\text{FEDformer}_t$ | $2.45 \pm 0.03$ | $5.43 \pm 0.04$ | $1.67 \pm 0.02$ | $3.22 \pm 0.04$ | $44.87 \pm 0.16$ | $66.54 \pm 0.07$ |
| $\text{STWA}_t$ | $1.98 \pm 0.02$ | $3.51 \pm 0.04$ | $1.64 \pm 0.01$ | $3.04 \pm 0.02$ | $45.28 \pm 0.13$ | $68.93 \pm 0.04$ |
| $\text{MTGNN}_t$ | $\underline{1.93 \pm 0.01}$ | $\underline{3.35 \pm 0.03}$ | $\underline{1.54 \pm 0.01}$ | $\underline{2.96 \pm 0.02}$ | $\underline{43.32 \pm 0.02}$ | $\underline{65.81 \pm 0.07}$ |
| BiTGraph | $\mathbf{1.56 \pm 0.02}$ | $\mathbf{2.97 \pm 0.02}$ | $\mathbf{1.51 \pm 0.01}$ | $\mathbf{2.92 \pm 0.02}$ | $\mathbf{42.11 \pm 0.11}$ | $\mathbf{65.53 \pm 0.23}$ |

| Method | PEMS | | ETTh1 | | BeijingAir | |
|---|---|---|---|---|---|---|
| (0.8) | MAE | RMSE | MAE | RMSE | MAE | RMSE |
| BRITS | $3.26 \pm 0.10$ | $7.02 \pm 0.03$ | $2.14 \pm 0.01$ | $3.81 \pm 0.09$ | $46.75 \pm 0.59$ | $68.17 \pm 0.87$ |
| SPIN | $2.26 \pm 0.01$ | $5.03 \pm 0.02$ | – | – | $\mathbf{44.94 \pm 0.13}$ | $\mathbf{66.58 \pm 0.25}$ |
| GRIN | $2.96 \pm 0.02$ | $6.70 \pm 0.12$ | – | – | $52.07 \pm 0.92$ | $74.60 \pm 1.85$ |
| GCN-M | $2.54 \pm 0.02$ | $5.77 \pm 0.04$ | – | – | $52.57 \pm 0.08$ | $73.71 \pm 0.14$ |
| CRUs | $3.15 \pm 0.01$ | $6.07 \pm 0.01$ | $3.15 \pm 0.11$ | $5.31 \pm 0.09$ | $57.10 \pm 1.45$ | $73.18 \pm 0.14$ |
| $\text{AGCRN}_0$ | $5.09 \pm 0.00$ | $10.07 \pm 0.02$ | $3.18 \pm 0.12$ | $6.02 \pm 0.02$ | $55.59 \pm 0.03$ | $81.58 \pm 0.10$ |
| $\text{Transformer}_0$ | $2.97 \pm 0.06$ | $6.73 \pm 0.08$ | $2.81 \pm 0.07$ | $5.20 \pm 0.06$ | $52.47 \pm 0.03$ | $73.10 \pm 0.16$ |
| $\text{FEDformer}_0$ | $2.89 \pm 0.04$ | $6.37 \pm 0.06$ | $2.59 \pm 0.03$ | $5.78 \pm 0.02$ | $56.94 \pm 0.04$ | $81.88 \pm 0.09$ |
| $\text{STWA}_0$ | $2.34 \pm 0.04$ | $5.16 \pm 0.02$ | $2.56 \pm 0.02$ | $4.41 \pm 0.03$ | $52.07 \pm 0.11$ | $74.95 \pm 0.08$ |
| $\text{MTGNN}_0$ | $2.39 \pm 0.02$ | $5.18 \pm 0.09$ | $2.46 \pm 0.04$ | $5.30 \pm 0.02$ | $53.05 \pm 0.02$ | $73.57 \pm 0.10$ |
| $\text{AGCRN}_t$ | $5.10 \pm 0.01$ | $10.07 \pm 0.01$ | $3.16 \pm 0.09$ | $5.25 \pm 0.12$ | $55.48 \pm 0.05$ | $78.86 \pm 0.13$ |
| $\text{Transformer}_t$ | $2.85 \pm 0.07$ | $6.29 \pm 0.11$ | $2.51 \pm 0.01$ | $4.98 \pm 0.07$ | $50.09 \pm 0.06$ | $72.00 \pm 0.07$ |
| $\text{FEDformer}_t$ | $2.76 \pm 0.03$ | $6.18 \pm 0.02$ | $2.55 \pm 0.02$ | $4.69 \pm 0.06$ | $49.83 \pm 0.15$ | $71.22 \pm 0.08$ |
| $\text{STWA}_t$ | $2.27 \pm 0.02$ | $5.00 \pm 0.01$ | $2.11 \pm 0.02$ | $3.73 \pm 0.04$ | $46.97 \pm 0.06$ | $72.93 \pm 0.07$ |
| $\text{MTGNN}_t$ | $\underline{2.21 \pm 0.01}$ | $\underline{4.89 \pm 0.01}$ | $\underline{2.01 \pm 0.01}$ | $3.73 \pm 0.02$ | $45.93 \pm 0.04$ | $68.17 \pm 0.11$ |
| BiTGraph | $\mathbf{2.15 \pm 0.01}$ | $\mathbf{4.73 \pm 0.02}$ | $\mathbf{1.91 \pm 0.01}$ | $\mathbf{3.54 \pm 0.01}$ | $\underline{45.47 \pm 0.17}$ | $66.98 \pm 0.29$ |

Table 6: The results of forecasting error on PEMS, ETTh1, and Beijing Air datasets with the missing rate $r = 0.2$, 0.4, and 0.6.

| Method (0.2) | PEMS MAE | PEMS RMSE | ETTh1 MAE | ETTh1 RMSE | BeijingAir MAE | BeijingAir RMSE |
|---|---|---|---|---|---|---|
| BRITS | $3.07 \pm 0.02$ | $6.34 \pm 0.00$ | $1.81 \pm 0.05$ | $3.24 \pm 0.00$ | $47.81 \pm 0.16$ | $70.66 \pm 2.41$ |
| SPIN | $2.08 \pm 0.05$ | $4.74 \pm 0.16$ | – | – | $\underline{44.03 \pm 0.12}$ | $\mathbf{66.34 \pm 0.75}$ |
| GRIN | $2.69 \pm 0.09$ | $6.21 \pm 0.22$ | – | – | $46.58 \pm 0.03$ | $68.81 \pm 1.37$ |
| GCN-M | $2.18 \pm 0.01$ | $5.07 \pm 0.06$ | – | – | $50.35 \pm 0.03$ | $70.51 \pm 0.08$ |
| CRUs | $2.82 \pm 0.01$ | $5.60 \pm 0.03$ | $2.87 \pm 0.05$ | $4.92 \pm 0.05$ | $57.81 \pm 0.55$ | $75.49 \pm 0.61$ |
| $AGCRN_0$ | $5.10 \pm 0.03$ | $10.04 \pm 0.01$ | $2.57 \pm 0.02$ | $4.93 \pm 0.05$ | $55.47 \pm 0.16$ | $81.06 \pm 0.12$ |
| $Transformer_0$ | $2.79 \pm 0.04$ | $6.20 \pm 0.07$ | $2.31 \pm 0.07$ | $3.79 \pm 0.04$ | $50.37 \pm 0.06$ | $73.18 \pm 0.07$ |
| $FEDformer_0$ | $2.70 \pm 0.06$ | $5.71 \pm 0.13$ | $1.80 \pm 0.03$ | $3.36 \pm 0.02$ | $50.03 \pm 0.09$ | $71.81 \pm 0.14$ |
| $STWA_0$ | $2.07 \pm 0.01$ | $4.69 \pm 0.03$ | $1.82 \pm 0.01$ | $3.23 \pm 0.03$ | $45.16 \pm 0.05$ | $67.96 \pm 0.13$ |
| $MTGNN_0$ | $2.10 \pm 0.03$ | $4.63 \pm 0.02$ | $1.65 \pm 0.02$ | $3.07 \pm 0.02$ | $44.71 \pm 0.09$ | $66.11 \pm 0.06$ |
| $AGCRN_t$ | $5.10 \pm 0.00$ | $10.05 \pm 0.01$ | $2.35 \pm 0.02$ | $4.17 \pm 0.04$ | $53.67 \pm 0.08$ | $78.55 \pm 0.12$ |
| $Transformer_t$ | $2.62 \pm 0.01$ | $5.42 \pm 0.02$ | $1.75 \pm 0.03$ | $3.30 \pm 0.06$ | $47.72 \pm 0.04$ | $69.17 \pm 0.16$ |
| $FEDformer_t$ | $2.56 \pm 0.03$ | $5.77 \pm 0.04$ | $1.72 \pm 0.01$ | $3.23 \pm 0.03$ | $45.70 \pm 0.09$ | $68.91 \pm 0.02$ |
| $STWA_t$ | $2.03 \pm 0.02$ | $4.61 \pm 0.07$ | $1.67 \pm 0.02$ | $3.12 \pm 0.01$ | $45.08 \pm 0.02$ | $67.57 \pm 0.10$ |
| $MTGNN_t$ | $\underline{1.98 \pm 0.01}$ | $\underline{4.51 \pm 0.06}$ | $\mathbf{1.56 \pm 0.01}$ | $\underline{3.05 \pm 0.01}$ | $44.40 \pm 0.06$ | $66.86 \pm 0.11$ |
| BiTGraph | $\mathbf{1.90 \pm 0.01}$ | $\mathbf{4.28 \pm 0.01}$ | $\mathbf{1.56 \pm 0.01}$ | $\mathbf{2.97 \pm 0.02}$ | $\mathbf{42.94 \pm 0.13}$ | $\underline{66.36 \pm 0.25}$ |

| Method (0.4) | PEMS MAE | PEMS RMSE | ETTh1 MAE | ETTh1 RMSE | BeijingAir MAE | BeijingAir RMSE |
|---|---|---|---|---|---|---|
| BRITS | $3.08 \pm 0.01$ | $6.31 \pm 0.03$ | $1.82 \pm 0.07$ | $3.26 \pm 0.06$ | $46.20 \pm 0.35$ | $67.95 \pm 0.06$ |
| SPIN | $2.13 \pm 0.10$ | $4.81 \pm 0.24$ | – | – | $45.37 \pm 0.55$ | $67.38 \pm 0.77$ |
| GRIN | $2.81 \pm 0.03$ | $6.76 \pm 0.14$ | – | – | $46.79 \pm 0.25$ | $68.22 \pm 0.14$ |
| GCN-M | $2.26 \pm 0.02$ | $5.51 \pm 0.03$ | – | – | $50.61 \pm 0.04$ | $71.61 \pm 0.17$ |
| CRUs | $2.85 \pm 0.02$ | $5.65 \pm 0.04$ | $2.94 \pm 0.05$ | $5.01 \pm 0.04$ | $55.80 \pm 0.46$ | $76.41 \pm 1.16$ |
| $AGCRN_0$ | $5.09 \pm 0.01$ | $10.05 \pm 0.02$ | $3.19 \pm 0.03$ | $5.06 \pm 0.07$ | $55.52 \pm 0.07$ | $81.27 \pm 0.03$ |
| $Transformer_0$ | $2.86 \pm 0.02$ | $6.38 \pm 0.07$ | $2.36 \pm 0.04$ | $4.79 \pm 0.08$ | $53.11 \pm 0.06$ | $77.03 \pm 0.17$ |
| $FEDformer_0$ | $2.81 \pm 0.05$ | $5.81 \pm 0.09$ | $1.96 \pm 0.02$ | $3.61 \pm 0.03$ | $52.18 \pm 0.05$ | $74.78 \pm 0.16$ |
| $STWA_0$ | $2.12 \pm 0.02$ | $4.67 \pm 0.03$ | $1.90 \pm 0.03$ | $3.40 \pm 0.03$ | $47.38 \pm 0.07$ | $71.16 \pm 0.06$ |
| $MTGNN_0$ | $2.12 \pm 0.01$ | $\underline{4.50 \pm 0.02}$ | $1.92 \pm 0.02$ | $3.43 \pm 0.01$ | $45.17 \pm 0.08$ | $66.73 \pm 0.02$ |
| $AGCRN_t$ | $5.10 \pm 0.00$ | $10.06 \pm 0.01$ | $2.87 \pm 0.04$ | $5.24 \pm 0.03$ | $55.56 \pm 0.07$ | $82.67 \pm 0.10$ |
| $Transformer_t$ | $2.80 \pm 0.03$ | $6.09 \pm 0.07$ | $1.88 \pm 0.02$ | $3.68 \pm 0.08$ | $48.28 \pm 0.16$ | $69.71 \pm 0.09$ |
| $FEDformer_t$ | $2.74 \pm 0.02$ | $5.78 \pm 0.03$ | $1.86 \pm 0.02$ | $3.31 \pm 0.01$ | $46.96 \pm 0.13$ | $68.17 \pm 0.08$ |
| $STWA_t$ | $2.07 \pm 0.03$ | $4.80 \pm 0.02$ | $1.81 \pm 0.04$ | $3.26 \pm 0.07$ | $45.69 \pm 0.13$ | $70.56 \pm 0.19$ |
| $MTGNN_t$ | $\underline{2.05 \pm 0.02}$ | $4.61 \pm 0.07$ | $\underline{1.67 \pm 0.01}$ | $\underline{3.12 \pm 0.01}$ | $\underline{44.29 \pm 0.02}$ | $\underline{66.39 \pm 0.08}$ |
| BiTGraph | $\mathbf{1.96 \pm 0.00}$ | $\mathbf{4.34 \pm 0.02}$ | $\mathbf{1.64 \pm 0.02}$ | $\mathbf{3.07 \pm 0.02}$ | $\mathbf{43.13 \pm 0.22}$ | $\mathbf{65.55 \pm 0.24}$ |

| Method (0.6) | PEMS MAE | PEMS RMSE | ETTh1 MAE | ETTh1 RMSE | BeijingAir MAE | BeijingAir RMSE |
|---|---|---|---|---|---|---|
| BRITS | $3.14 \pm 0.03$ | $6.22 \pm 0.01$ | $1.87 \pm 0.11$ | $\underline{3.24 \pm 0.02}$ | $46.71 \pm 0.17$ | $67.93 \pm 0.04$ |
| SPIN | $2.18 \pm 0.05$ | $4.91 \pm 0.15$ | – | – | $\underline{44.28 \pm 0.45}$ | $\underline{65.68 \pm 0.08}$ |
| GRIN | $2.84 \pm 0.01$ | $6.61 \pm 0.1$ | – | – | $49.10 \pm 1.01$ | $71.43 \pm 1.63$ |
| GCN-M | $2.39 \pm 0.02$ | $5.32 \pm 0.03$ | – | – | $51.47 \pm 0.05$ | $77.41 \pm 0.22$ |
| CRUs | $2.97 \pm 0.31$ | $5.79 \pm 0.09$ | $3.05 \pm 0.05$ | $5.13 \pm 0.04$ | $56.63 \pm 0.11$ | $75.42 \pm 1.76$ |
| $AGCRN_0$ | $5.09 \pm 0.01$ | $10.06 \pm 0.01$ | $3.43 \pm 0.02$ | $5.81 \pm 0.05$ | $55.67 \pm 0.06$ | $82.10 \pm 0.05$ |
| $Transformer_0$ | $2.91 \pm 0.03$ | $6.31 \pm 0.04$ | $2.45 \pm 0.03$ | $4.30 \pm 0.03$ | $50.79 \pm 0.11$ | $71.73 \pm 0.09$ |
| $FEDformer_0$ | $2.85 \pm 0.04$ | $6.02 \pm 0.06$ | $2.06 \pm 0.01$ | $3.96 \pm 0.02$ | $55.32 \pm 0.13$ | $79.83 \pm 0.03$ |
| $STWA_0$ | $2.15 \pm 0.02$ | $4.64 \pm 0.06$ | $2.16 \pm 0.02$ | $4.86 \pm 0.09$ | $48.22 \pm 0.07$ | $70.09 \pm 0.15$ |
| $MTGNN_0$ | $2.13 \pm 0.01$ | $4.62 \pm 0.02$ | $2.36 \pm 0.02$ | $4.14 \pm 0.03$ | $47.05 \pm 0.11$ | $67.80 \pm 0.06$ |
| $AGCRN_t$ | $5.10 \pm 0.00$ | $10.06 \pm 0.01$ | $3.35 \pm 0.02$ | $5.29 \pm 0.04$ | $55.45 \pm 0.17$ | $80.76 \pm 0.04$ |
| $Transformer_t$ | $2.83 \pm 0.04$ | $5.22 \pm 0.07$ | $2.19 \pm 0.01$ | $4.29 \pm 0.03$ | $49.23 \pm 0.04$ | $72.38 \pm 0.12$ |
| $FEDformer_t$ | $2.79 \pm 0.02$ | $6.31 \pm 0.05$ | $2.07 \pm 0.01$ | $4.26 \pm 0.03$ | $49.21 \pm 0.06$ | $68.27 \pm 0.07$ |
| $STWA_t$ | $2.14 \pm 0.02$ | $4.67 \pm 0.03$ | $1.85 \pm 0.02$ | $3.37 \pm 0.03$ | $46.06 \pm 0.03$ | $69.92 \pm 0.07$ |
| $MTGNN_t$ | $\underline{2.11 \pm 0.01}$ | $\underline{4.59 \pm 0.03}$ | $\underline{1.79 \pm 0.02}$ | $3.29 \pm 0.03$ | $44.85 \pm 0.02$ | $67.78 \pm 0.05$ |
| BiTGraph | $\mathbf{1.99 \pm 0.01}$ | $\mathbf{4.47 \pm 0.01}$ | $\mathbf{1.74 \pm 0.00}$ | $\mathbf{3.21 \pm 0.01}$ | $\mathbf{44.23 \pm 0.15}$ | $\mathbf{64.20 \pm 0.32}$ |

Table 7: The results of forecasting error on Metr-LA and ETTh1 datasets with the block missing rate $r = 0.0015$ and $0.002$.

| Method | Metr-LA | | | ETTh1 | | |
|---|---|---|---|---|---|---|
| ($r = 0.0015$) | MAE | RMSE | MAPE | MAE | RMSE | MAPE |
| BRITS | $8.66 \pm 0.02$ | $12.24 \pm 0.02$ | $18.49 \pm 0.03$ | $2.12 \pm 0.01$ | $4.55 \pm 0.04$ | $96.73 \pm 0.09$ |
| SPIN | $6.58 \pm 0.02$ | $11.02 \pm 0.03$ | $12.76 \pm 0.03$ | – | – | – |
| GRIN | $6.73 \pm 0.02$ | $11.27 \pm 0.03$ | $12.92 \pm 0.04$ | – | – | – |
| GCN-M | $6.77 \pm 0.03$ | $11.68 \pm 0.02$ | $13.06 \pm 0.03$ | – | – | – |
| CRU | $7.32 \pm 0.02$ | $12.04 \pm 0.03$ | $14.86 \pm 0.02$ | $1.92 \pm 0.00$ | $4.08 \pm 0.02$ | $81.09 \pm 0.06$ |
| $\text{AGCRN}_0$ | $14.88 \pm 0.01$ | $14.30 \pm 0.02$ | $29.94 \pm 0.02$ | $2.79 \pm 0.00$ | $5.02 \pm 0.01$ | $116.64 \pm 0.13$ |
| $\text{Transformer}_0$ | $7.59 \pm 0.02$ | $12.87 \pm 0.03$ | $16.03 \pm 0.02$ | $2.04 \pm 0.01$ | $4.20 \pm 0.02$ | $95.09 \pm 0.05$ |
| $\text{FEDformer}_0$ | $7.50 \pm 0.03$ | $12.55 \pm 0.01$ | $15.94 \pm 0.03$ | $2.01 \pm 0.00$ | $3.96 \pm 0.02$ | $85.28 \pm 0.04$ |
| $\text{STWA}_0$ | $7.12 \pm 0.01$ | $14.84 \pm 0.03$ | $13.62 \pm 0.03$ | $1.98 \pm 0.01$ | $3.69 \pm 0.02$ | $79.29 \pm 0.09$ |
| $\text{MTGNN}_0$ | $7.18 \pm 0.03$ | $15.49 \pm 0.04$ | $18.31 \pm 0.02$ | $1.92 \pm 0.01$ | $3.76 \pm 0.01$ | $81.47 \pm 0.06$ |
| $\text{AGCRN}_t$ | $14.88 \pm 0.00$ | $14.30 \pm 0.01$ | $29.94 \pm 0.02$ | $2.50 \pm 0.00$ | $4.92 \pm 0.02$ | $107.79 \pm 0.12$ |
| $\text{Transformer}_t$ | $7.08 \pm 0.01$ | $11.75 \pm 0.03$ | $13.24 \pm 0.04$ | $1.81 \pm 0.02$ | $3.39 \pm 0.01$ | $84.67 \pm 0.06$ |
| $\text{FEDformer}_t$ | $6.81 \pm 0.02$ | $11.56 \pm 0.03$ | $13.15 \pm 0.03$ | $1.77 \pm 0.00$ | $3.43 \pm 0.01$ | $73.58 \pm 0.07$ |
| $\text{STWA}_t$ | $6.56 \pm 0.01$ | $\mathbf{10.57 \pm 0.04}$ | $12.90 \pm 0.03$ | $1.79 \pm 0.01$ | $3.21 \pm 0.02$ | $74.59 \pm 0.06$ |
| $\text{MTGNN}_t$ | $\underline{6.41 \pm 0.01}$ | $10.99 \pm 0.02$ | $\underline{12.74 \pm 0.04}$ | $\underline{1.61 \pm 0.00}$ | $\underline{3.09 \pm 0.02}$ | $\underline{68.31 \pm 0.07}$ |
| BiTGraph | $\mathbf{6.22 \pm 0.01}$ | $\underline{10.97 \pm 0.03}$ | $\mathbf{12.70 \pm 0.02}$ | $\mathbf{1.56 \pm 0.00}$ | $\mathbf{3.03 \pm 0.01}$ | $\mathbf{66.20 \pm 0.03}$ |
| Method | Metr-LA | | | ETTh1 | | |
| ($r = 0.002$) | MAE | RMSE | MAPE | MAE | RMSE | MAPE |
| BRITS | $9.09 \pm 0.02$ | $11.94 \pm 0.04$ | $18.39 \pm 0.11$ | $2.37 \pm 0.02$ | $4.66 \pm 0.03$ | $107.38 \pm 0.06$ |
| SPIN | $6.68 \pm 0.03$ | $11.19 \pm 0.02$ | $12.88 \pm 0.04$ | – | – | – |
| GRIN | $6.87 \pm 0.02$ | $12.18 \pm 0.06$ | $13.47 \pm 0.03$ | – | – | – |
| GCN-M | $6.85 \pm 0.02$ | $11.74 \pm 0.03$ | $13.35 \pm 0.03$ | – | – | – |
| CRU | $7.91 \pm 0.02$ | $13.25 \pm 0.03$ | $17.63 \pm 0.05$ | $2.97 \pm 0.01$ | $5.05 \pm 0.00$ | $97.58 \pm 0.11$ |
| $\text{AGCRN}_0$ | $14.88 \pm 0.00$ | $14.26 \pm 0.02$ | $29.90 \pm 0.09$ | $2.82 \pm 0.01$ | $5.13 \pm 0.04$ | $124.61 \pm 0.13$ |
| $\text{Transformer}_0$ | $8.13 \pm 0.02$ | $13.67 \pm 0.02$ | $18.05 \pm 0.04$ | $2.19 \pm 0.00$ | $4.24 \pm 0.01$ | $103.02 \pm 0.08$ |
| $\text{FEDformer}_0$ | $8.02 \pm 0.01$ | $13.36 \pm 0.03$ | $17.93 \pm 0.06$ | $2.13 \pm 0.01$ | $4.20 \pm 0.01$ | $85.07 \pm 0.05$ |
| $\text{STWA}_0$ | $7.60 \pm 0.01$ | $16.76 \pm 0.07$ | $16.38 \pm 0.03$ | $2.00 \pm 0.00$ | $3.76 \pm 0.00$ | $79.43 \pm 0.04$ |
| $\text{MTGNN}_0$ | $7.36 \pm 0.02$ | $11.51 \pm 0.09$ | $15.28 \pm 0.03$ | $2.06 \pm 0.01$ | $4.08 \pm 0.02$ | $79.84 \pm 0.04$ |
| $\text{AGCRN}_t$ | $14.88 \pm 0.01$ | $14.26 \pm 0.02$ | $29.90 \pm 0.02$ | $2.43 \pm 0.01$ | $4.82 \pm 0.00$ | $99.99 \pm 0.16$ |
| $\text{Transformer}_t$ | $7.47 \pm 0.03$ | $12.34 \pm 0.04$ | $15.72 \pm 0.03$ | $1.76 \pm 0.00$ | $3.35 \pm 0.01$ | $75.32 \pm 0.08$ |
| $\text{FEDformer}_t$ | $7.42 \pm 0.02$ | $12.11 \pm 0.06$ | $15.54 \pm 0.03$ | $1.85 \pm 0.01$ | $3.45 \pm 0.01$ | $87.11 \pm 0.07$ |
| $\text{STWA}_t$ | $6.65 \pm 0.01$ | $\underline{10.92 \pm 0.03}$ | $12.72 \pm 0.03$ | $1.79 \pm 0.00$ | $3.33 \pm 0.02$ | $76.30 \pm 0.10$ |
| $\text{MTGNN}_t$ | $\underline{6.49 \pm 0.02}$ | $11.20 \pm 0.01$ | $\underline{12.60 \pm 0.04}$ | $\underline{1.68 \pm 0.01}$ | $\underline{3.27 \pm 0.02}$ | $\underline{71.41 \pm 0.11}$ |
| BiTGraph | $\mathbf{6.31 \pm 0.01}$ | $\mathbf{10.84 \pm 0.02}$ | $\mathbf{12.49 \pm 0.02}$ | $\mathbf{1.60 \pm 0.01}$ | $\mathbf{3.18 \pm 0.02}$ | $\mathbf{68.11 \pm 0.04}$ |

## B MORE COMPARISON

Table 4 presents the results of different methods for the missing rate $r = 0.1$ on Metr-LA and Electricity datasets. Table 5 and Table 6 demonstrates the results on ETTh1, PEMS and BeijingAir datasets. Our proposed BiTGraph demonstrates significant superiority, especially in the Electricity dataset with the highest variance. The performance of $\text{STWA}_t$ is very close to that of $\text{MTGNN}_t$, and both models outperform the other baseline models. Moreover, the BiTGraph improves the MAE by $5.12$ percent under the largest mask ratio on the PEMS dataset, which further demonstrates the effectiveness of our proposed method.

## C BLOCK MISSING

To further substantiate the effectiveness of our method, we assess its performance in block-missing scenarios. We follow the method of GRIN to generate block-missing masks, and the results are presented in Table 7. We can observe that the proposed BiTGraph also demonstrates its great superiority, achieving a notable 5% improvement over the best baseline.

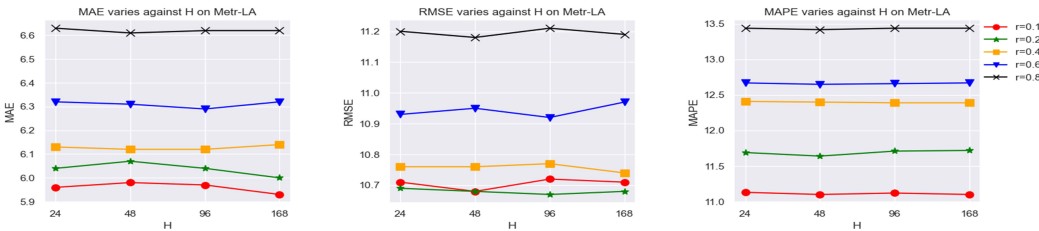

Figure 2: The performance under different window sizes with $H$=24, 48, 96, and 168

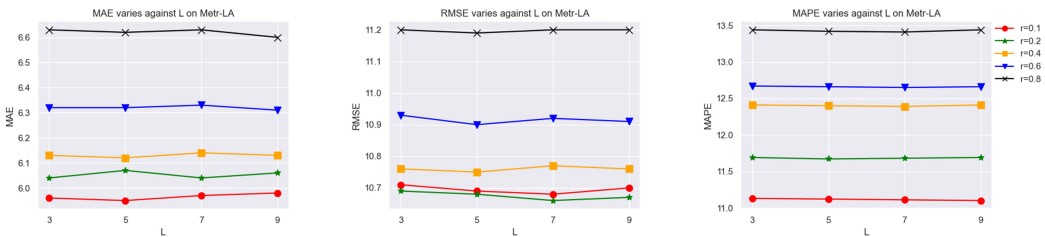

Figure 3: The performance under $L$=3, 5, 7, and 9

## D   Hyperparameter Sensitivity

In this section, we evaluate the hyperparameter sensitivity of our method with respect to the history window size $H$, the number of blocks $L$, as well as the number of nearest neighbors $k$ on the Metr-LA dataset.

Figure 2 shows the performance changes over the history window size $H$. No evident performance improvements are observed as $H$ increases from 24 to 168. We then evaluate the sensitivity of $L$. As shown in Figure 3, BiTGraph achieves the best performance when $L$=5. However, the performance does not improve further when $L$ continues growing. We hypothesize that this is due to the over-smoothing issue of GNNs, that is, the node representations become indistinguishable when the graph convolutional layer reaches 5. We then further assess the impact of the $k$ on the model performance. In particular, We study the metrics change as $k$ varies from 5, 10, 15, to 20. As observed from Figure 4, the performance gains by incorporating spatial information from more neighboring nodes. However, once enough spatial correlations are obtained (i.e., $k$=10), further increasing the number of neighbors will no longer yield performance improvement. This suggests that only a few neighborhood nodes have a significant impact on a given node.

## E   The role of $\beta$

Table 8 shows the change of learnable $\beta$ against missing rates on three representative datasets (traffic, solar energy, and air quality). We found that $\beta$ demonstrates similar values with different missing rates, indicating that $\beta$ is primarily responsible for adjusting the strength of correctness and is

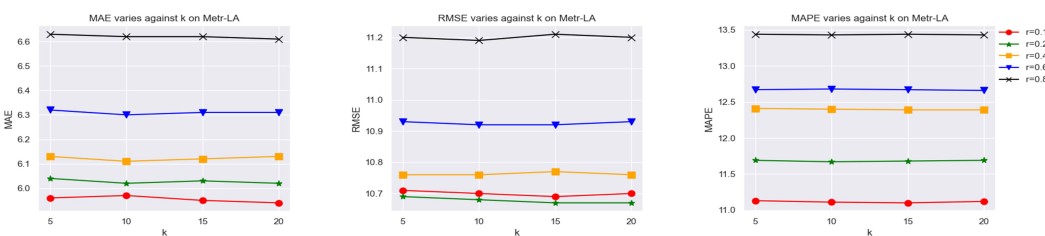

Figure 4: The performance under $k$ =5, 10, 15, and 20

Table 8: The learned value of $\beta$ on three datasets.

| Dataset | 0.2 | 0.4 | 0.6 | 0.8 |
|---|---|---|---|---|
| Metr-LA | 0.035 | 0.037 | 0.034 | 0.035 |
| BeijingAir | 0.475 | 0.463 | 0.471 | 0.467 |
| ETTh1 | 0.397 | 0.391 | 0.419 | 0.419 |

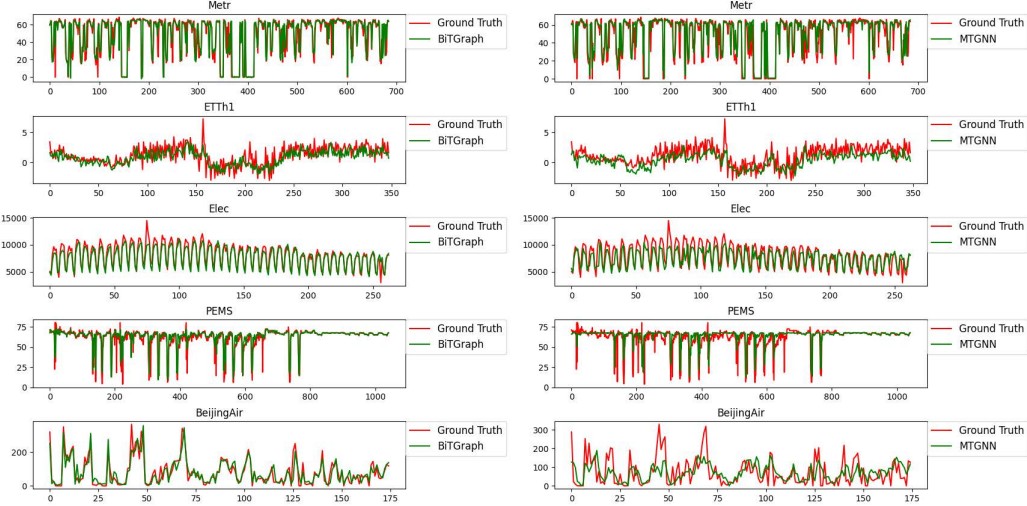

Figure 5: The forecasting curves produced by our BiTGraph on four datasets.

independent of missing rates. Additionally, the possible reason for the value variation of $\beta$ across different datasets may caused by the unique characteristics of different datasets.

## F  PREDICTION VISUALIZATION

In this section, we demonstrate the forecasting curves of our method on four datasets with a missing ratio $r = 0.8$. The time series is down-sampled in the temporal domain, and it is randomly selected in the spatial domain. As shown in the left part of Figure 5, our proposed method is able to produce results that well match the ground truth trends on all five datasets, which demonstrates the effectiveness of BiTGraph. In addition, we present the forecasting results of one representative method $MTGNN_t$, which are displayed in the right part of Figure 5. We can observe that the curves produced by $MTGNN_t$ struggle to match the ground truths when the missing rate is high. Especially, on the BeijingAir dataset, the forecasting curves largely deviate from the ground truths.

## G  MODEL COMPLEXITY ANALYSIS

In this section, we assess the complexity of different methods in terms of memory usage and number of parameters. Table 9 shows the model complexity of different methods. It can be observed that our proposed BiTGraph has a relatively small memory usage as well as a small number of parameters, which implies the model can be trained with much less energy consumption.

Table 9: The model complexity of different methods.

| Method | Memory Usage | #Parameters |
|---|---|---|
| BRITS | 85.48M | 173.72K |
| SPIN | 10.85G | 1.31K |
| GRIN | 2.00G | 12.76K |
| GCN-M | 15.39G | 396.50K |
| CRUs | 41.97M | 54.75K |
| AGCRN | 10.17M | 1.56K |
| MTGNN | 250.55M | 9.184K |
| Transformer | 14.85G | 15.46M |
| FEDformer | 15.57G | 16.08M |
| STWA | 1.96G | 256.50K |
| BiTGraph | 194.17M | 11.82K |

