# OpenReview forum: "Biased Temporal Convolution Graph Network for Time Series Forecasting with Missing Values"
_ICLR.cc/2024/Conference — ICLR 2024 poster_

### Official Review · Reviewer_dZwn · 2023-10-27

**Soundness:** 3 good
**Presentation:** 3 good
**Contribution:** 3 good
**Rating:** 6
**Confidence:** 4

**Summary:**

To handle missing values in time series, the paper employs
partial convolutions from computer vision. The authors then
use a graph neural network with channels as nodes and
a induced adjacency matrix for forecasting. They evaluate
their model on several datasets and show that it ourperforms
several baselines from the literature.

**Strengths:**

s1. handling missing values is an important problem,
   esp. in time series.
s2. treating missing values with partial convolutions is interesting.
s3. the method consistently reduces errors in the experiments.
s4. ablation study shows effect of the two different components
  of the method.

**Weaknesses:**

w1. limited methodological contribution: the paper merely combines
  two existing methods.
w2.  missing principled  baseline: forecast based on the time series
  **and the imputation mask**.

**Questions:**

Using partial convolutions is a simple and plausible approach to
treat missing values in time series (s2). The experiments show a
consistent decrease in error over five datasets (s3) and the ablation
study shows the effect of the two different components (s4).

Two points one could discuss:
w1. limited methodological contribution: the paper merely combines
  two existing methods.

w2.  missing principled  baseline: forecast based on the time series
  **and the imputation mask**.
In the main experiment in table 2 the authors compare several
forecasting models for completely observed time series after
imputing zeros ("Transformer_0") and after imputing with a
specific imputation method ("Transformer_t"). However, this way
the information which observations have been missing gets lost,
and this might be relevant ("informed missingness").  The default
approach is to impute a zero **and** to add a channel carrying
the imputation mask (i.e., as "1" if the value was observed and
a "0" if it originally was missing and now has been imputed).
This way, the information about missingness is not lost. -- It
would be really important to add these principled baselines
to make sure that the specific way the proposed models deals
with the missing values really is causing the observed differences.

---

> ### Author Response · Authors · 2023-11-23
> **Response to Reviewer dZwn**
>
> Q1
>
> > Limited methodological contribution: the paper merely combines two existing methods.
>
> A1
>
> We would like to **highlight the innovations** of our proposed model from three aspects.
> - For the PartialTCN module, we introduce the **instance PartialTCN** that focuses on capturing the temporal dependencies whereas it leaves the instance correlation for Biased GCN, this stands in contrast to the PartialCNN which relies on the convolution to capture the correlation from the two dimensions, and thus allowing for the **exploration of temporal patterns that are invariant to instances**.
> - For the Biased GCN module, we innovate the GCN by **adding a bias term (prior knowledge) to account for different missing patterns** (illustrated by Equation-9 in the revised manuscript), which will then be used to **correct the message passing strength** in the graph information diffusion. Our ablation study shows that such innovation is critical to the forecasting performance.
> - We stack multiple blocks of the two modules and design a hierarchical architecture, and propose **a new mask updating mechanism** in both modules (as shown in Equation-6 and Equation-10) to **ensure a corrected mask** when information flows from bottom to up.
>
> We have **updated both the content and Figure-1 (b)** in the revised manuscript to highlight these innovations.
>
> Q2
>
> > missing principled baseline: to impute a zero and to add a channel carrying the imputation mask (i.e., as "1" if the value was observed and a "0" if it originally was missing and now has been imputed). This way, the information about missingness is not lost.
>
> A2
>
> Thanks for your suggestion. We have **changed the experimental setting to make the imputation mask available to all baseline methods**. This can indeed improve the performance of most methods, especially for those Transformer-based ones. We have **revised the manuscript** to include the corresponding analysis and discussion.

---

### Official Review · Reviewer_naqz · 2023-11-01

**Soundness:** 3 good
**Presentation:** 4 excellent
**Contribution:** 3 good
**Rating:** 6
**Confidence:** 4

**Summary:**

This paper introduces a Biased Temporal Convolution Graph Network (BiaTCGNet) for forecasting from partially observed time series. BiaTCGNet is designed to jointly capture the temporal dependencies within and spatial structure of time series, while accounting for missing patterns by injecting bias into its components (MultiScale Instance PartialTCN and Biased GCN). Experiments conducted on several real-world benchmarks demonstrate the effectiveness of BiaTCGNet over alternative approaches.

**Strengths:**

* The proposed Biased Temporal Convolution Graph Network (BiaTCGNet) is designed specifically to account for both the temporal and spatial aspects of multivariate time series. Namely, this is achieved through BiaTCGNet’s two constituent modules: a Multi-Scale Instance PartialTCN and a Biased GCN. The former is capable of performing instance-independent temporal convolution to capture temporal (intra-instance) correlations within each individual time series, while the latter constructs a graph and diffuses information over it to capture the spatial correlations between the time series instances (channels).

* In contrast to other existing time series forecasting methods, BiaTCGNet explicitly considers missing values in its model design through bias injection in the Multi-Scale Instance PartialTCN module that helps account for the different missing patterns; while progressively updating the missing patterns during Biased GCN’s information diffusion process.

* Experiments on five real-world benchmark datasets have been conducted, the results of which suggest that BiaTCGNet achieves improvements of up to 11% over the existing forecasting methods under various scenarios involving missing values.

* The paper is technically sound, well written and organized in a reasonably clear and comprehensive manner. The notation used throughout the paper is clear and consistent.

Update: Thanks to the authors for detailed responses, after reading them, as well as comments from all reviewers, I am updating my rating from 5 to 6.

**Weaknesses:**

* Generally speaking, BiaTCGNet appears to be a result of (1) an almost direct application of a TCN with a straightforward modification to account for partial observations (Liu et al., 2018), and (2) leveraging a conventional GCN for learning of an adjacency matrix relying on two node embedding sets so as to capture asymmetric spatial correlations. In that regard, the novelty of this work may be considered incremental. Therefore, I would encourage the authors to further clarify and/or elaborate on the novelty of the two modules within BiaTCGNet.

* In Eq. (8), the authors introduce $\beta$, a learnable weight that is aimed to serve as a time-window-specific bias that corrects the global spatial correlation strength in accordance with the present missing patterns. Nevertheless, the role of this weight has not been discussed further. I believe that this work would benefit from including the learned values of $\beta$ for each individual dataset used in the experiments. If possible, I would also suggest that the authors consider including a brief discussion on the interpretation of those values.

* There seems to be a fairly recent work [C1] on attention-based memory networks for joint modeling of local spatio-temporal features and global historical patterns in multivariate time series with missing values. Moreover, the forecasting problem formulation in [C1] seems to be consistent with the one considered in this work. Therefore, I would suggest that the authors consider comparing the GCN-M method from [C1] with BiaTCGNet. Alternatively, I would ask the authors to provide the specific reason as to why this method has not been included among the baselines?

[C1] Zuo, J., Zeitouni, K., Taher, Y., & Garcia-Rodriguez, S. (2023). Graph convolutional networks for traffic forecasting with missing values. Data Mining and Knowledge Discovery, 37(2), 913-947.

**Questions:**

My questions and suggestions for the authors are included along with the weaknesses of this work (in the “Weaknesses” section of this review).

---

> ### Author Response · Authors · 2023-11-23
> **Response to Reviewer naqz**
>
> Q1
>
> >I would encourage the authors to further clarify and/or elaborate on the novelty of the two modules within BiaTCGNet.
>
> A1
>
> We would like to **highlight the innovations** of our proposed model from three aspects.
> - For the PartialTCN module, we introduce the **instance PartialTCN** that focuses on capturing the temporal dependencies whereas it leaves the instance correlation for Biased GCN, this stands in contrast to the PartialCNN which relies on the convolution to capture the correlation from the two dimensions, and thus allowing for the **exploration of temporal patterns that are invariant to instances**.
> - For the Biased GCN module, we innovate the GCN by **adding a bias term (prior knowledge) to account for different missing patterns** (illustrated by Equation-9 in the revised manuscript), which will then be used to **correct the message passing strength** in the graph information diffusion. Our ablation study shows that such innovation is critical to the forecasting performance.
> - We stack multiple blocks of the two modules and design a hierarchical architecture, and propose **a new mask updating mechanism** in both modules (as shown in Equation-6 and Equation-10) to **ensure a corrected mask** when information flows from bottom to up.
>
> We have **updated both the content and Figure-1 (b)** in the revised manuscript to highlight these innovations.
>
> Q2
> > The role of $\beta$ has not been discussed further. If possible, I would also suggest that the authors consider including a brief discussion on the interpretation of those values.
>
> A2
>
> Thanks for your suggestion. The following table shows the change of learnable $\beta$ against missing rates on three representative datasets (traffic, solar energy, and air quality). We found that **$\beta$ demonstrates similar values with different missing rates**, indicating that $\beta$ is primarily responsible for adjusting the strength of correctness and is independent of missing rates. Additionally, the possible reason for **the value variation of $\beta$ across different datasets** may caused by the unique characteristics of different datasets. The discussion can be found in Section E of the Appendix.
>
> Table R2-1(The values of $\beta$)
> |**Dataset**| **0.2** | **0.4** | **0.6** |**0.8** |
> |------|------|------|------|------|
> |Metr-LA|0.035|0.037|0.034|0.035|
> |BeijingAir|0.475|0.463|0.471|0.467|
> |ETTh1|0.397|0.391|0.419|0.419|
>
> Q3
>
> >There seems to be a fairly recent work [C1] on attention-based memory networks for joint modeling of local spatio-temporal features and global historical patterns in multivariate time series with missing values. Moreover, the forecasting problem formulation in [C1] seems to be consistent with the one considered in this work. Therefore, I would suggest that the authors consider comparing the GCN-M method from [C1] with BiaTCGNet. Alternatively, I would ask the authors to provide the specific reason as to why this method has not been included among the baselines?
>
> A3
>
> Thanks for pointing us the missing work. The GCN-M [1] is now **included as a baseline method** in our revised manuscript and its results are available in **Section 5.3 and Sections B, C, and G**. Note that as GCN-M requires location information, it is only applicable to the Metr-LA, PEMS, and Beijing Air datasets.
>
> References:
>
> [1] Jingwei Zuo, Karine Zeitouni, Yehia Taher, and Sandra Garcia-Rodriguez. Graph convolutional networks for traffic forecasting with missing values. Data Mining and Knowledge Discovery, 2023.

---

### Official Review · Reviewer_sYzZ · 2023-11-03

**Soundness:** 2 fair
**Presentation:** 4 excellent
**Contribution:** 3 good
**Rating:** 6
**Confidence:** 3

**Summary:**

The paper presented a multivariate time series forecasting model that applies temporal convolution and graph convolution. Such model is shown to handle missing values in inputs without additional pre-processing effort and outperforms the tested approaches.

**Strengths:**

The idea of introducing masked temporal convolution from vision tasks to time series is quite interesting and novel.
The paper conducted intensive experiments including comparison against multiple methods and ablation studies.
The paper is well written in the technical details.

**Weaknesses:**

The experiment setup is not clear to me whether it's sound.
Specifically, (1) It is not clear what loss function is used in both the proposed method and the tested methods. The paper compared MAE, MAPE and RMSE; it's the authors' choice what loss function to be used. Hence, if the task is to optimize for RMSE, the loss function for all the tested methods should be RMSE because the output would be the optimal mean estimator. Without such clarity, it's hard to derive where the accuracy difference comes from; is it from the matching loss function and evaluation metric, or is it from architecture innovation?
(2) for MAE and MAPE, both are the evaluation metric for the median estimate as the denominator from MAPE, without clarifying, is assumed to be the same for all methods? But I saw for example in Table 2, under these two metrics, the ranking of methods can differ. This made me worry whether the experiment setup or the metric calculation is different from what I assumed.

Another minor note: an interval is implied for all the reported results and only in caption from Table 2, it was mentioned that 'the results are averaged over 5 runs'. What are these 5 runs referring to? Do they refer to different random seeds for training only? I assumed so given that the paper mentioned the train/test/validate is splitter based on ratio chronically which will fix the dataset so no variation from the data.

**Questions:**

Introducing missing masks to feature map, motivated from vision tasks, makes sense; yet it'd be good to compare against, introducing the missing masks as input feature directly so the model can be trained with the knowledge what inputs are missing.

It was implied that the proposed approach could capture missing pattern; yet the experiments seem to only tested the scenario where values are missing at random. In actual application, it's rare that values are missing at random. It'd be good to also test different scenarios to show the efficacy of the proposed methods.

---

> ### Author Response · Authors · 2023-11-23
> **Response to Reviewer sYzZ**
>
> Q1
>
> > The experiment setup is not clear to me whether it's sound. Is it from the matching loss function and evaluation metric, or is it from architecture innovation.
>
> >Under the rank of MAE and MAPE, the ranking of methods can differ.
>
> A1
>
> Thanks for pointing us this ambiguity. We actually adopt the **MAE** as our **training loss**, and we have revised the Equation-2 to make it clear.  Regarding the ranking difference of MAE and MAPE, they are indeed the same metric up to a normalization (denominator), however, the **denominators are not a fixed constant** and change against different points (y_t). As a consequence, the **ranking of the two metrics can differ** even for the same method and dataset, and this is also observed in prior work [1, 2].
>
> Q2
>
> > It was mentioned that 'the results are averaged over 5 runs'. What are these 5 runs referring to? Do they refer to different random seeds for training only?
>
> A2
>
> Yes, the five-runs mean **different random seeds for training only**, i.e., we run a method for 5 rounds with different random seeds and calculate the averaged values from the 5 rounds. We have revised the paper to make it clear now.
>
> Q3
>
> > Introducing missing masks to feature map, motivated from vision tasks, makes sense; yet it'd be good to compare against, introducing the missing masks as input feature directly so the model can be trained with the knowledge what inputs are missing.
>
> A3
>
> Thanks for your suggestion. We have **changed the experimental setting for all baseline methods** such that they all take the missing masks as direct input features on all datasets. This can indeed improve the performance of most methods, especially for those Transformer-based ones. We have **revised the manuscript** to include the corresponding analysis and discussion.
>
> Q4
>
> > It was implied that the proposed approach could capture missing pattern; yet the experiments seem to only tested the scenario where values are missing at random. In actual application, it's rare that values are missing at random. It'd be good to also test different scenarios to show the efficacy of the proposed methods.
>
> A4
>
> Thanks for your suggestion. We now included an **additional experiment** to verify the effectiveness of our proposed method **under the block missing scenarios**. In particular, we follow the setting of prior work [3] by adopting two block-missing rates of 0.15\% and 0.2\% to test different models on the Metr-LA and ETTh1 datasets. The corresponding results are presented in Table 4 of the Appendix. Notably, our proposed **BiaTCGNet** demonstrates its superiority consistently, **achieving a remarkable 5\% improvement** over the best baseline method.
>
> References:
>
> [1] Zonghan Wu, Shirui Pan, Guodong Long, Jing Jiang, Xiaojun Chang, and Chengqi Zhang. Connecting the dots: Multivariate time series forecasting with graph neural networks. In ACM SIGKDD Conference on Knowledge Discovery and Data Mining (SIGKDD), 2020.
>
> [2] Jingwei Zuo, Karine Zeitouni, Yehia Taher, and Sandra Garcia-Rodriguez. Graph convolutional networks for traffic forecasting with missing values. Data Mining and Knowledge Discovery, 2023.
>
> [3] Andrea Cini, Ivan Marisca, and Cesare Alippi. Filling the gaps: Multivariate time series imputation by graph neural networks. In International Conference on Learning Representations (ICLR), 2022.

---

### Author Response · Authors · 2023-11-23
**General Response To All Reviewers**

We sincerely thank all reviewers for all the precious comments and valuable advice. In view of these suggestions, we conducted several additional experiments and revised our article accordingly. Our major changes are summarized as follows.

- We provided **additional clarification on the novelty of the two modules** within BiaTCGNet in both Introduction (Section 1) and Methodology (Section 4).
- We **changed the experimental setting for all baseline methods** such that they all take the missing masks as direct input features on all datasets.
- We **included GCN-M model as a baseline method** in our experiments, and the corresponding analysis and discussion are also presented.
- We **added the block missing patterns** to further verify the effectiveness of our proposed BiaTCGNet and the corresponding results are shown in Section C of the Appendix.
- We **provided explanations on the learned values of $\beta$** in Section E of the Appendix.

---

### Meta-Review · Area_Chair_cMyb · 2023-12-04

**Metareview:**

The reviewers were unanimous in their vote of Weak Accept. Quoting one of the reviews, "The idea of introducing masked temporal convolution from vision tasks to time series is quite interesting and novel. The paper conducted intensive experiments including comparison against multiple methods and ablation studies. The paper is well written in the technical details."

**Justification For Why Not Higher Score:**

Highest score was a 6 and concerns were raised as to whether the advance was incremental.

**Justification For Why Not Lower Score:**

All reviewers voted Weak Accept.

---

### Decision · Program_Chairs · 2024-01-16

Accept (poster)